# Denoising Autoregressive Transformers for Scalable Text-to-Image Generation

**Jiatao Gu[†], Yuyang Wang[†], Yizhe Zhang[†], Qihang Zhang[δ*],**
**Dinghuai Zhang[γ*], Navdeep Jaitly[†], Josh Susskind[†], Shuangfei Zhai[†]**
[†]Apple, [δ]The Chinese University of Hong Kong, [γ]Mila
[†]{jgu32,yuyangw,yizzhang,njaitly,jsusskind,szhai}@apple.com
[δ]qhzhang@link.cuhk.edu.hk [γ]dinghuai.zhang@mila.quebec

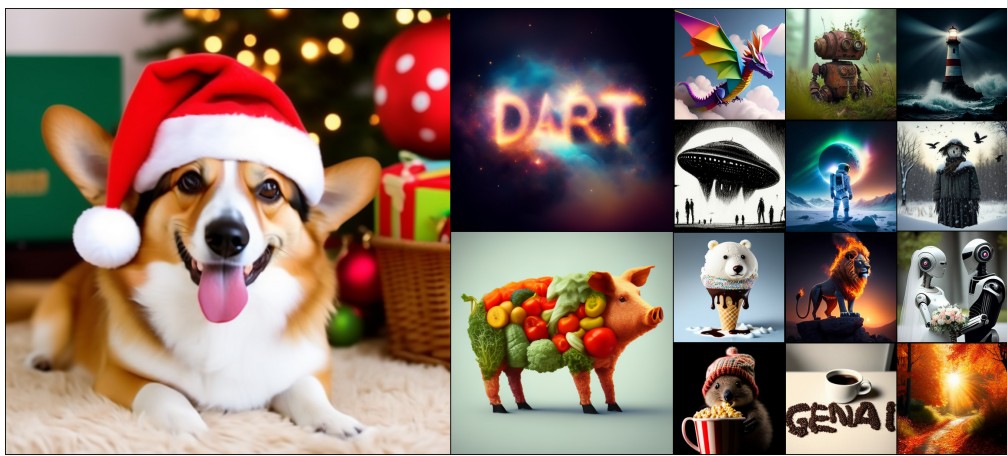

Figure 1: Curated examples of images generated by DART at $256^2$, $512^2$ and $1024^2$ pixels.

## Abstract

Diffusion models have become the dominant approach for visual generation. They are trained by denoising a Markovian process which gradually adds noise to the input. We argue that the Markovian property limits the model's ability to fully utilize the generation trajectory, leading to inefficiencies during training and inference. In this paper, we propose DART, a transformer-based model that unifies autoregressive (AR) and diffusion within a non-Markovian framework. DART iteratively denoises image patches spatially and spectrally using an AR model that has the same architecture as standard language models. DART does not rely on image quantization, which enables more effective image modeling while maintaining flexibility. Furthermore, DART seamlessly trains with both text and image data in a unified model. Our approach demonstrates competitive performance on class-conditioned and text-to-image generation tasks, offering a scalable, efficient alternative to traditional diffusion models. Through this unified framework, DART sets a new benchmark for scalable, high-quality image synthesis.

## 1 Introduction

Recent advancements in deep generative models have led to significant breakthroughs in visual synthesis, with diffusion models emerging as the dominant approach for generating high-quality images (Rombach et al., 2022; Esser et al., 2024). Diffusion models (Sohl-Dickstein et al., 2015; Ho et al., 2020) operate by progressively adding Gaussian noise to an image and learning to reverse this process in a sequence of denoising steps. Despite their success, these models are difficult to

---

*Work done as part of an internship at Apple.

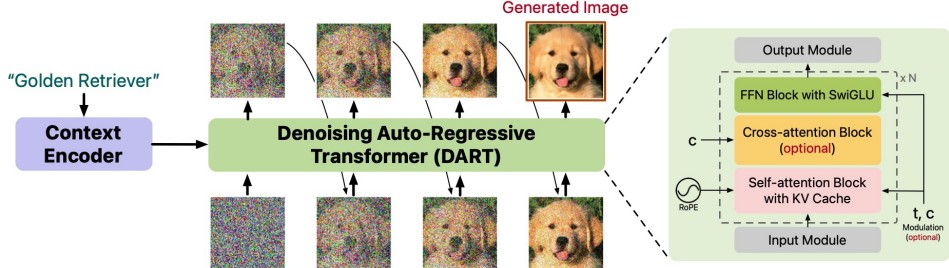

Figure 2: (←) A general illustration of the proposed DART. The model autoregressively denoises image through a Transformer until a clean image is generated. Here, whole images are shown for visualization purpose; (→) We show the architecture details which integrates state-of-the-art designs similar to common language models (Dubey et al., 2024).

train on high resolution images directly, requiring either cascaded models (Ho et al., 2022), or multiscale approaches (Gu et al., 2023) or preprocessing of images to autoencoder codes at lower resolutions (Rombach et al., 2022). These limitations can stem from their reliance on the Markovian assumption, which simplifies the generative process but restricts the model only to see the generation from the previous step. This often leads to inefficiencies during training and inference, as the individual steps of denoising are unaware of the trajectory of generations from prior steps.

In parallel, autoregressive models, such as GPT-4 (Achiam et al., 2023), have shown great success in modeling long-range dependencies in sequential data, particularly in the field of natural language processing. These models efficiently cache computations and manage dependencies across time steps, which has also inspired research into adapting autoregressive models for image generation. However, early efforts such as PixelCNN (Oord et al., 2016), while promising, suffered from high computational costs due to pixel-wise generation. More recent models like VQ-GAN (Esser et al., 2021a) and related work (Yu et al., 2022; Team, 2024; Tian et al., 2024) learn models of quantized images in a compressed latent space; Li et al. (2024) propose to generate directly in such space without quantization by employing a diffusion-based loss function. However, these methods fail to fully leverage the progressive denoising benefits of diffusion models, resulting in limited global context and error propagation during generation.

To address these limitations, we propose Denoising AutoRegressive Transformer (DART), a novel generative model that integrates autoregressive modeling within a non-Markovian diffusion framework (Song et al., 2021) (Fig. 2). The non-Markovian formulation in DART enables the model to leverage the full generative trajectory during training and inference, while retaining the progressive modeling benefits of diffusion models, resulting in more efficient and flexible generation compared to traditional diffusion and autoregressive approaches. Additionally, DART introduces two key improvements to address the limitations of the non-Markovian approach: (1) token-level autoregressive modeling (DART-AR), which captures dependencies between image tokens autoregressively, enabling finer control and improved generation quality, and (2) a flow-based refinement module (DART-FM), which enhances the model's expressiveness and smooths transitions between denoising steps. These extensions make DART a flexible and efficient framework capable of handling a wide range of tasks, including class conditional, text-to-image, as well as multimodal generation.

DART offers a scalable, efficient alternative to traditional diffusion models, achieving competitive performance on standard benchmarks for class-conditioned (e.g., ImageNet (Deng et al., 2009)) and text-to-image generation. To summarize, major contributions of our work include:

- We propose DART, a novel non-Markovian diffusion model that leverages the full denoising trajectory, leading to more efficient and flexible image generation compared to traditional approaches.

- We propose two key improvements: DART-AR and DART-FM, which improve the expressiveness and coherence throughout the non-Markovian generation process.

- DART achieves competitive performance in both class-conditioned and text-to-image generation tasks, offering a scalable and unified approach for high-quality, controllable image synthesis.

## 2 BACKGROUND

### 2.1 DIFFUSION MODELS

Diffusion models (Sohl-Dickstein et al., 2015; Ho et al., 2020; Song et al., 2020) are latent variable models given a pre-defined posterior distribution, and trained with a denoising objective. These models have gained widespread use in image generation (Rombach et al., 2021; Peebles & Xie, 2022; Podell et al., 2023; Esser et al., 2024). Diffusion models produce the entire image in a non-autoregressive manner through iterative processes. Specifically, given an image $\boldsymbol{x}_0 \in \mathbb{R}^{3 \times H \times W}$, we define a series of latent variables $\boldsymbol{x}_t$ ($t = 1, \cdots, T$) with a *Markovian process* which gradually adds noise to the original image $\boldsymbol{x}_0$. The transition $q(\boldsymbol{x}_t | \boldsymbol{x}_{t-1})$ and the marginal $q(\boldsymbol{x}_t | \boldsymbol{x}_0)$ probabilities are defined as follows, respectively:

$$q(\boldsymbol{x}_t | \boldsymbol{x}_{t-1}) = \mathcal{N}(\boldsymbol{x}_t; \sqrt{1 - \beta_t} \boldsymbol{x}_{t-1}, \beta_t \mathbf{I}), \quad q(\boldsymbol{x}_t | \boldsymbol{x}_0) = \mathcal{N}(\boldsymbol{x}_t; \sqrt{\bar{\alpha}_t} \boldsymbol{x}_0, (1 - \bar{\alpha}_t) \mathbf{I}), \quad (1)$$

where $\bar{\alpha}_t = \prod_{\tau=1}^{t}(1 - \beta_\tau), 0 < \beta_t < 1$ are determined by the noise schedule. The model learns to reverse this process with a backward model $p_\theta(\boldsymbol{x}_{t-1} | \boldsymbol{x}_t)$, which aims to denoise the image. The training objective for the model is:

$$\min \mathcal{L}_\theta^{\text{DM}} = \mathbb{E}_{t \sim [1, T], \boldsymbol{x}_t \sim q(\boldsymbol{x}_t | \boldsymbol{x}_0)}[\omega_t \cdot \|\boldsymbol{x}_\theta(\boldsymbol{x}_t, t) - \boldsymbol{x}_0\|_2^2], \quad (2)$$

where $\boldsymbol{x}_\theta(\boldsymbol{x}_t, t)$ is a time-conditioned denoiser that learns to map the noisy sample $\boldsymbol{x}_t$ to its clean version $\boldsymbol{x}_0$; $\omega_t$ is a time-dependent loss weighting, which usually uses SNR (Ho et al., 2020) or SNR+1 (Salimans & Ho, 2022). Practically, $\boldsymbol{x}_t$ can be re-parameterized with noise- or v-prediction (Salimans & Ho, 2022) for enhanced performance, and can be applied on pixel space (Gu et al., 2023; Saharia et al., 2022) or latent space, encoded by a VAE encoder (Rombach et al., 2021). However, standard diffusion models are computationally inefficient, requiring numerous denoising steps and extensive training data. Moreover, they lack the ability to leverage generation context effectively, hindering scalability to complex scenes and long sequences like videos.

### 2.2 AUTOREGRESSIVE MODELS

In the field of natural language processing, Transformer models have achieved notable success in autoregressive modeling (Vaswani et al., 2017; Raffel et al., 2020). Building on this success, similar approaches have been applied to image generation (Parmar et al., 2018; Esser et al., 2021a; Chen et al., 2020; Yu et al., 2022; Sun et al., 2024; Team, 2024). Different from diffusion-based methods, these methods typically focus on learning the dependencies among discrete image tokens (e.g., through Vector Quantization (Van Den Oord et al., 2017)). To elaborate, consider an image $\boldsymbol{x} \in \mathbb{R}^{3 \times H \times W}$. The process begins by encoding this image into a sequence of discrete tokens $\boldsymbol{z}_{1:N} = \mathcal{E}(\boldsymbol{x})$. These tokens are designed to approximately reconstruct the original image through a learned decoder $\hat{\boldsymbol{x}} = \mathcal{D}(\boldsymbol{z}_{1:N})$. An autoregressive model is then trained by maximizing the cross-entropy as follows:

$$\max \mathcal{L}_\theta^{\text{CE}} = \sum_{n=1}^{N} \log P_\theta(\boldsymbol{z}_n | \boldsymbol{z}_{0:n-1}), \quad (3)$$

where $\boldsymbol{z}_0$ is the special start token. During the inference phase, the autoregressive model is first used to sample tokens from the learned distribution, and then decode them into image space using $\mathcal{D}$.

As discussed in Kilian et al. (2024), autoregressive models offer significant efficiency advantages over diffusion models by caching previous steps in memory and enabling the entire generation process to be computed in a single parallel forward pass. This reduces computational overhead and accelerates training and inference. However, the reliance on quantization can lead to information loss, potentially degrading generation quality. Additionally, the linear, step-by-step nature of token prediction may overlook the global structure, making it challenging to capture long-range dependencies and holistic coherence in complex scenes or sequences.

## 3 DART

### 3.1 NON-MARKOVIAN DIFFUSION FORMULATION

We start by revisiting the basics of diffusion models from the perspective of hierarchical variational auto-encoders (HVAEs, Kingma & Welling, 2013; Child, 2021). Given a data-point $\boldsymbol{x}_0$, a hierarchical

VAE models generation process $p_\theta$ of a sequence of latent variables $\boldsymbol{x}_t (t = T \dots 1)$[1] by maximizing a evidence lower bound (ELBO):

$$\max \mathcal{L}_{\theta,\phi}^{\text{ELBO}} = \mathbb{E}_{\boldsymbol{x}_{1:T} \sim q_\phi(\boldsymbol{x})} \left[ \sum_{t=1}^T \log p_\theta(\boldsymbol{x}_{t-1} | \boldsymbol{x}_{t:T}) + \log p_\theta(\boldsymbol{x}_T) - \log q_\phi(\boldsymbol{x}_{1:T} | \boldsymbol{x}_0) \right], \quad (4)$$

where $\boldsymbol{x}_0$ is the real data, and $q_\phi$ is a learnable inference model. As pointed out in VDM (Kingma et al., 2021), diffusion models can essentially be seen as HVAEs with three specific modifications:

1. A fixed inference process $q$ which gradually adds noise to corrupt data $\boldsymbol{x}_0$;
2. Markovian forward and backward process where $\boldsymbol{x}_t$ depends only on $\boldsymbol{x}_{t+1}$ (Eq (1));
3. Noise-dependent loss weighting that reweighs ELBO with a focus on perception.

Only with all above simplifications combined, standard diffusion models can be formulated as Eq (2) where the generator becomes Markovian $p_\theta(\boldsymbol{x}_{t-1} | \boldsymbol{x}_{t:T}) = p_\theta(\boldsymbol{x}_{t-1} | \boldsymbol{x}_t)$ so that one can randomly sample $t$ to learn each transition independently. This greatly simplifies modeling, enabling training models with sufficiently large number of steps (e.g., $T = 1000$ for original DDPM (Ho et al., 2020)) without suffering from memory issues.

In prior research, these aspects are highly coupled, and few works attempt to disentangle them. We speculate that the Markovian assumption might not be a necessary requirement for a high generation quality, as long as the fixed posterior distribution and flexible loss weightings are maintained. As a side evidence, one can achieve reasonable generation with much fewer steps (e.g., 100) at inference time using non-Markovian HVAE. On the contrary, the Markovian modeling forces all information compressed solely in the corrupted data from previous noise level which could be an obstacle preventing efficient learning and require more inference steps.

**NOn-MArkovian Diffusion Models (NOMAD)** In this paper, we reconsider the original form of generator $p_\theta(\boldsymbol{x}_{t-1} | \boldsymbol{x}_{t:T})$ of HVAEs while maintaining the modifications, 1 and 3, made by diffusion models. More precisely, we learn the following weighted ELBO loss (Eq (4)) with adjustable $\tilde{\omega}_t$:

$$\max \mathcal{L}_\theta^{\text{NOMAD}} = \mathbb{E}_{\boldsymbol{x}_{1:T} \sim q(\boldsymbol{x}_0)} \left[ \sum_{t=1}^T \tilde{\omega}_t \cdot \log p_\theta(\boldsymbol{x}_{t-1} | \boldsymbol{x}_{t:T}) \right], \quad (5)$$

where $q$ is the pre-defined inference process. This formulation shares many similarities as autoregressive (AR) models in Eq (3), where in our case, each token represents a noisy sample $\boldsymbol{x}_t$. Therefore, it is natural to implement such process with autoregressive Transformers (Vaswani et al., 2017).

However, the Markovian inference process of standard diffusion models makes it impossible for the generator $\theta$ to use the entire context except for $\boldsymbol{x}_t$. That is to say, even if we initiate our generator as $p_\theta(\boldsymbol{x}_{t-1} | \boldsymbol{x}_{t:T})$, the model only needs information in $\boldsymbol{x}_t$ in order to best denoise $\boldsymbol{x}_{t-1}$. Therefore, it is critical to design a **fixed** and **non-Markovian**[2] inference process $q(\boldsymbol{x}_t | \boldsymbol{x}_{0:t-1})$ to sample the noisy sequences $\boldsymbol{x}_{1:T} \sim \boldsymbol{x}_0$. The simplest approach is to perform an *independent* noising process:

$$q(\boldsymbol{x}_t | \boldsymbol{x}_{0:t-1}) = q(\boldsymbol{x}_t | \boldsymbol{x}_0) = \mathcal{N}(\boldsymbol{x}_t; \sqrt{\gamma_t} \boldsymbol{x}_0, (1 - \gamma_t) \mathbf{I}), \quad \forall t \in [1, T], \quad (6)$$

where $\gamma_t$ represents the non-Markovian noise schedule. Note that, while Eq (6) may look close to the marginal distribution of the original diffusion models (Eq (1)), the underlying meaning is different as $\boldsymbol{x}_t$ is conditionally independent given $\boldsymbol{x}_0$. In practice, one can also choose a more complex non-Markovian process as presented in DDIM (Song et al., 2021), and we leave this exploration in future work. Additionally, we can show the following proposition:

**Proposition 1.** *A non-Markovian diffusion process $\{\boldsymbol{x}_t\}_t$ with independent noising $\gamma_t$ has a bijection to a Markovian diffusion process $\{\boldsymbol{y}_t\}_t$ with the same number of steps and noise level $\{\bar{\alpha}_t\}_t$; $\{\boldsymbol{y}_t\}_t$ achieves the maximal signal-to-noise ratio when its noise level satisfies $\frac{\bar{\alpha}_t}{1 - \bar{\alpha}_t} = \sum_{\tau=t}^T \frac{\gamma_\tau}{1 - \gamma_\tau}$.*

We defer the proof to Appendix A. The above proposition indicates that we can carefully choose a set of noise level $\gamma_t$ to model the same information-destroying process as a comparable diffusion baseline, while modeling with the entire generation trajectory.

---

[1] We follow the same notation of time indexing in diffusion models for consistency.

[2] The term non-Markovian refers to $\boldsymbol{x}_t$ is not only related to the previous step $\boldsymbol{x}_{t-1}$.

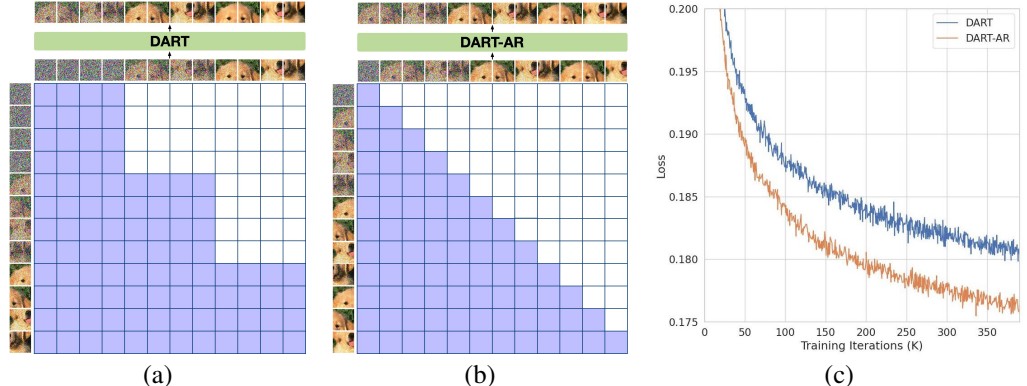

(a)                            (b)                            (c)

Figure 3: Attention masks for (a) DART and (b) DART-AR, highlighting their different structures. (c) Comparison of learning curves, demonstrating the superior performance of DART-AR.

## 3.2 PROPOSED METHODS

**Denoising AutoRegressive Transformer (DART)**   We propose DART – a Transformer-based generative model that implements non-Markovian diffusion with an independent noising process (see Figure 2). First, following DiT (Peebles & Xie, 2022), we represent each image by first extracting the latent map with a pretrained VAE (Rombach et al., 2021), patchify, and flatten the map into a sequence of continuous tokens $\boldsymbol{x}_t \in \mathbb{R}^{K \times C}$, where $K$ is the length, and $C$ is the channel dimension. When considering multiple noise levels, we concat tokens along the length dimension. Then, DART models the generation as $p_\theta(\boldsymbol{x}_{t-1}|\boldsymbol{x}_{t:T}) = \mathcal{N}\left(\boldsymbol{x}_{t-1}; \sqrt{\gamma_{t-1}}\boldsymbol{x}_\theta(\boldsymbol{x}_{t:T}), (1 - \gamma_{t-1})\mathbf{I}\right)$, where $\boldsymbol{x}_\theta(.)$ is a Transformer network that takes in the concatenated sequence $\boldsymbol{x}_{t:T} \in \mathbb{R}^{K(T-t) \times C}$, and predicts the "mean" of the next noisy image. By combining with Eq (6), we simplify Eq (5) as:

$$\min \mathcal{L}_\theta^{\text{DART}} = \mathbb{E}_{\boldsymbol{x}_{1:T} \sim q(\boldsymbol{x}_0)} \left[ \sum_{t=1}^{T} \omega_t \cdot \|\boldsymbol{x}_\theta(\boldsymbol{x}_{t:T}) - \boldsymbol{x}_0\|_2^2 \right], \tag{7}$$

where we define $\omega_t = \frac{\gamma_{t-1}}{1-\gamma_{t-1}}\tilde{\omega}_t$ to simplify the notation. Similar to standard AR models, training of $T$ denoising steps is in parallel, where computations across different steps are shared. A chunk-based causal mask is used to maintain the autoregressive structure (see Figure 3(a)).

It is evident from Eq (7) that the objective is similar to the original diffusion objective (Eq (2)), demonstrating that DART can be trained as robustly as standard diffusion models. Additionally, by leveraging the diffusion trajectory within an autoregressive framework, DART allows us to incorporate the proven design principles of large language models (Brown et al., 2020; Dubey et al., 2024). Furthermore, Proposition 1 indicates that we can select $\omega_t$ according to its associated diffusion process. For instance, with SNR weighting (Ho et al., 2020), $\omega_t$ can be defined as $\omega_t = \sum_{\tau=t}^{T} \frac{\gamma_\tau}{1-\gamma_\tau}$.

Sampling from DART is straightforward: we simply predict the mean $\boldsymbol{x}_\theta(\boldsymbol{x}_{t:T})$, add Gaussian noise to obtain the next step $\hat{\boldsymbol{x}}_{t-1}$, and feed that to the following iteration. Unlike diffusion models, no complex solvers are needed. Similar to diffusion models, classifier-free guidance (CFG, Ho & Salimans, 2021) is applied to the prediction of $\boldsymbol{x}_\theta$ for improved visual quality. Additionally, KV-cache is employed to enhance decoding efficiency.

**Limitations of Naive DART**   Unlike diffusion models, which can be trained with a large number of steps $T$, non-Markovian modeling is constrained by memory consumption as $T$ increases. For example, using $T = 16$ on $256 \times 256$ images will easily create over 4000 tokens even in the latent space. This fundamentally limits the modeling capacity on complex tasks such as text-to-image generation. However, the flexibility of the autoregressive structure allows us to enhance the capacity without compromising scalability. In this work, we propose two methods on top of DART:

**1. DART with Token Autoregressive (DART-AR)**   As similarly discussed by Xiao et al. (2021), the independent Gaussian assumption of $p_\theta(\boldsymbol{x}_{t-1}|\boldsymbol{x}_{t:T})$ is inaccurate to approximate the complex true

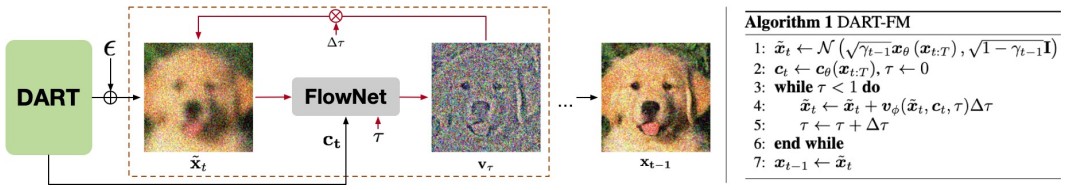

Figure 4: An illustration for the generation process of DART-FM.

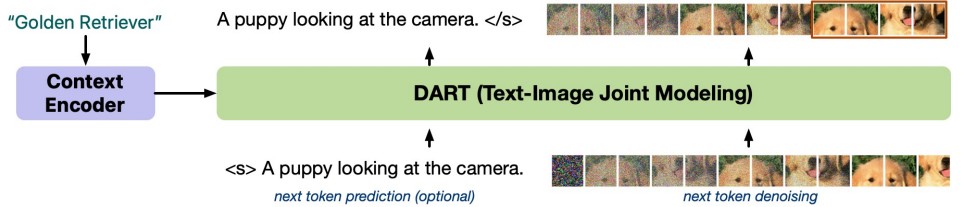

Figure 5: Illustrations of Kaleido-DART, an application for multi-modal generation.

distribution of $q(\boldsymbol{x}_{t-1}|\boldsymbol{x}_{t:T})$, especially when $T$ is small. A straightforward solution to model image denoising as an additional autoregressive model $p_\theta(\boldsymbol{x}_{t-1}|\boldsymbol{x}_{t:T}) = \prod_{k=1}^{K} p_\theta(\boldsymbol{x}_{t-1}^k|\boldsymbol{x}_{t-1}^{<k}, \boldsymbol{x}_{t:T})$:

$$\min \mathcal{L}_\theta^{\text{DART-AR}} = \mathbb{E}_{\boldsymbol{x}_{1:T} \sim q(\boldsymbol{x}_0)} \left[ \sum_{t=1}^{T} \omega_t \sum_{k=1}^{K} \|\boldsymbol{x}_\theta(\boldsymbol{x}_{t-1}^{<k}, \boldsymbol{x}_{t:T}) - \boldsymbol{x}_0^k\|_2^2 \right], \quad (8)$$

where $\boldsymbol{x}_{t-1}^1, \ldots, \boldsymbol{x}_{t-1}^K$ are the $K$ flatten tokens of $\boldsymbol{x}_{t-1}$. The autoregressive decomposition ensures each tokens are not independent, which is strictly stronger than the original DART. We demonstrate this by visualizing the training curve in Figure 3 (c). Training of DART-AR takes essentially the same amount of computation as standard DART with two additional modifications at the input and attention masks (see comparison in Figure 3 (a) (b)). At sampling time, DART-AR is relatively much more expensive as it requires $K \times T$ AR steps before it outputs the final prediction.

**2. DART with Flow Matching (DART-FM)**   The above approach models dependency across tokens, while maintaining Gaussian modeling at each step. Alternatively, we can improve the expressiveness of $p_\theta(\boldsymbol{x}_{t-1}|\boldsymbol{x}_{t:T})$ by abandoning the Gaussian assumption, similar to Li et al. (2024). More precisely, we first sample $\tilde{\boldsymbol{x}}_t$ as in regular DART. Then, we recursively apply a continuous flow network, $\boldsymbol{v}_\phi(\tilde{\boldsymbol{x}}_t, \boldsymbol{c}_t, \tau_t)$, over multiple iterations to bridge the gap between $\tilde{\boldsymbol{x}}_t$ and $\boldsymbol{x}_{t-1}$ (see Figure 4). Here, $\boldsymbol{v}_\phi(\tilde{\boldsymbol{x}}_t, \boldsymbol{c}_t, \tau)$ models the velocity field for the probability flow between the distributions of $\tilde{\boldsymbol{x}}_t$ and $\boldsymbol{x}_{t-1}$, $\tau \in [0, 1]$ denotes the auxiliary flow timestep, and $\boldsymbol{c}_t = \boldsymbol{c}_\theta(\boldsymbol{x}_{t:T})$ represents the features from the last Transformer block, providing contextual information across the noisy image. Consequently, a simple MLP suffices to model $\boldsymbol{v}_\phi$, adding only a minimal overhead to the total training cost. We train $\boldsymbol{v}_\phi$ via flow matching (Liu et al., 2022; Lipman et al., 2023; Albergo et al., 2023) due to its simplicity:

$$\min \mathcal{L}_{\phi,\theta}^{\text{FM}} = \mathbb{E}_{\boldsymbol{x}_{1:T} \sim q(\boldsymbol{x}_0)} \sum_{t=1}^{T} \mathbb{E}_{\tau \in [0,1]} \|\boldsymbol{v}_\phi((1-\tau)\tilde{\boldsymbol{x}}_t + \tau \boldsymbol{x}_{t-1}, \boldsymbol{c}_t, \tau) - (\boldsymbol{x}_{t-1} - \tilde{\boldsymbol{x}}_t)\|_2^2, \quad (9)$$

where $\tilde{\boldsymbol{x}}_t = \sqrt{\gamma_{t-1}}\text{SG}\left[\boldsymbol{x}_\theta(\boldsymbol{x}_{t:T})\right] + \sqrt{1 - \gamma_{t-1}}\epsilon$, and $\epsilon \sim \mathcal{N}(0, \mathbf{I})$. $\text{SG}\left[\right]$ is the stop-gradient operator to avoid trivial solutions in optimization. In practice, we combine Eq (9) with the original DART objectives, which can be seen as an additional refinement on top of the Gaussian-based prediction.

### 3.3   MULTI-MODAL GENERATION

Our proposed framework, built on an autoregressive model, naturally extends to discrete token modeling tasks. This includes discrete latent modeling for image generation (Gu et al., 2024) and multi-modal generation (Team, 2024). By leveraging a shared architecture, we jointly optimize for both continuous denoising and next-token prediction loss using cross-entropy for discrete latents

which we call Kaleido-DART considering its architectual similarity as Gu et al. (2024) (see Figure 5). To balance inference across modalities, we reweight the discrete loss (Eq (3)) according to the relative lengths between the discrete and image tokens:

$$\mathcal{L}_\theta^{\text{Kaleido}} = \lambda \mathcal{L}_\theta^{\text{CE}} + \mathcal{L}_\theta^{\text{DART}}, \qquad (10)$$

where $\lambda = \frac{\text{\# text tokens}}{\text{\# image tokens}}$. It is important to note that our approach is markedly distinct from several concurrent works aimed at unifying autoregressive and diffusion models within a single parameter space (Zhou et al., 2024; Xie et al., 2024; Zhao et al., 2024; Xiao et al., 2024). In these efforts, the primary goal is to adapt language model architectures to perform diffusion tasks, without modifying the underlying diffusion process itself to account for the shift in model design. As we have discussed, the Markovian nature of diffusion models inherently limits their ability to leverage generation history, a feature that lies at the core of autoregressive models. In contrast, DART is designed to merge the advantages of both autoregressive and diffusion frameworks, fully exploiting the autoregressive capabilities. This formulation also allows for seamless integration into LLM pipelines.

# 4 EXPERIMENTS

## 4.1 EXPERIMENTAL SETTINGS

**Dataset**   We experiment with DART on both class-conditioned image generation on ImageNet (Deng et al., 2009) and text-to-image generation on CC12M (Changpinyo et al., 2021), where each image is accompanied by a descriptive caption. All models are trained to synthesize images at $256 \times 256$. For multimodal generation tasks, we augment CC12M with synthetic captions as additional ground-truth.

**Evaluation**   In line with prior works, we report Fréchet Inception Distance (FID) (Heusel et al., 2017) to quantify the the realism and diversity of generated images. For text-to-image generation, we also use the CLIP score (Hessel et al., 2021) to measure how well the generated images align with the given text instructions. To assess the zero-shot capabilities of the models, we report scores based on the MSCOCO 2017 (Lin et al., 2014) validation set.

**Architecture**   Following § 3.2, we experimented with three variants of the proposed model: the default DART, along with two enhanced versions, DART-AR and DART-FM. All variants are implemented using the same Transformer blocks for consistency. As illustrated in Figure 2, our design is similar to Dubey et al. (2024), incorporating rotary positional encodings (RoPE, Su et al., 2024) within the self-attention layers and SwiGLU activation (Shazeer, 2020) in the FFN layers. For class-conditioned generation, we follow (Peebles & Xie, 2022), adding an AdaLN block to each Transformer block to integrate class-label information. For text-to-image generation, we replace AdaLN with additional cross-attention layers over pretrained T5-XL encoder (Raffel et al., 2020). Since DART uses a fixed noise schedule, there is no need for extra time embeddings as long as RoPE is active. In addition, for DART-FM, we incorporate a small flow network, implemented as a 3-layer MLP, which increases the total parameter count by only about 1%.

**Training**   Proposition 1 not only sets a connection between DART and standard diffusion models, but also allows us to define the noise schedule based on any existing diffusion schedule $\{\bar{\alpha}_t\}_t$, which can be inversely mapped to the DART schedule $\{\gamma_t\}_t$ using the bijection. In this paper, we adopt the cosine schedule $\bar{\alpha}_t = \cos(\pi/2 \cdot t/T)$. We set $T = 16$ while $K = 256^3$ throughout all experiments unless otherwise specified. We train all models with a batch size of 128 images, resulting in a total of 0.5M image tokens per update. We use the AdamW optimizer (Loshchilov & Hutter, 2017) with a cosine learning rate schedule, setting the maximum learning rate to 3e-4.

## 4.2 RESULTS

**Class-conditioned Generation**   We report the FID scores for conditional ImageNet generation in Figure 7(a), following the approach of previous works. In line with Li et al. (2024), we apply a linear CFG scheduler to DART and its variants. As shown in the figure, both -AR and -FM variants consistently outperform the default DART across all guidance scales, demonstrating the effectiveness

---

[3] $K = 256$ from encoding $256 \times 256$ images with StableDiffusion v1.4 VAE (https://huggingface.co/stabilityai/sd-vae-ft-ema) with a patch size of 2 and $C = 16$ channels.

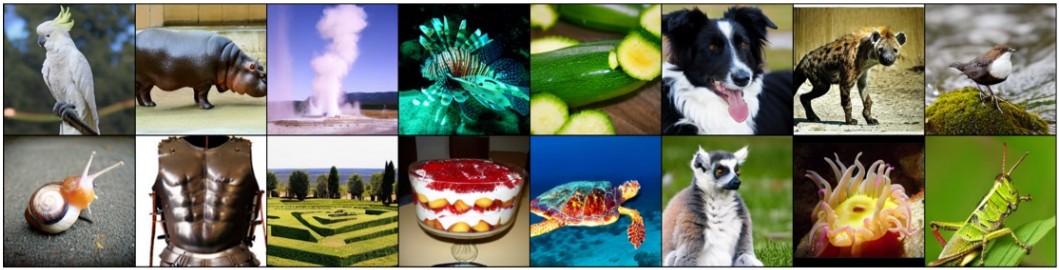

(a) Random samples of DART-AR trained on Class-conditioned Image Generation

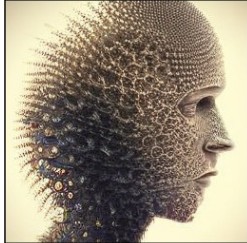
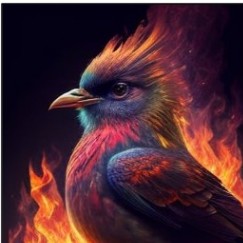
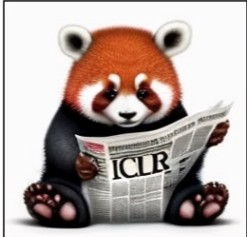
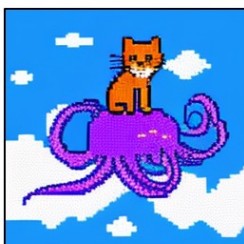

| human life depicted entirely out of fractals | This dreamlike digital art captures a vibrant, kaleidoscopic bird with fire in the background | a fluffy red panda reading a newspaper with ICLR written on it | Pixel art a cat riding a octopus through the clouds in the sky |

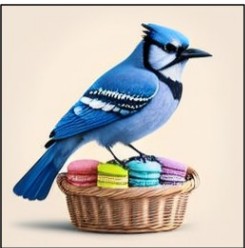
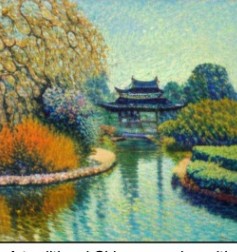
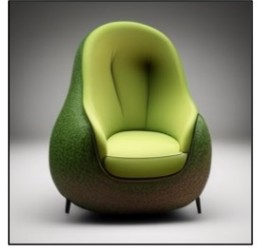
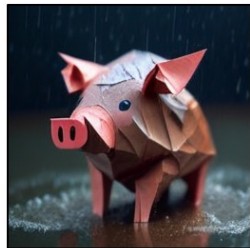

| A blue jay standing on a large basket of rainbow macarons | A traditional Chinese garden with river, an Impressionist painting | An armchair in the shape of an avocado | an origami pig all wet by heavy rain |

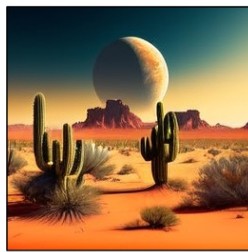
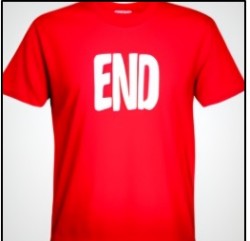
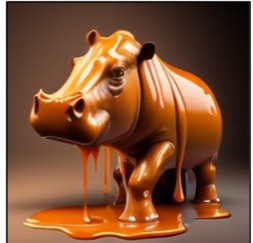
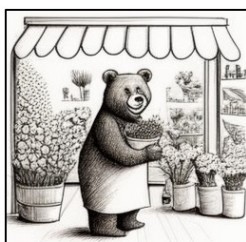

| A desert landscape on an alien planet, giant cacti, and a gas giant low on the horizon | the word 'End' on a red T-shirt | A hippo composed entirely of thick, flowing caramel, with sticky drips oozing from its body. Parts of the caramel stretch and melt. | Detailed pen and ink drawing of a happy bear lady selling flowers in her shop. |

(b) Selected samples of DART-FM trained on Text-to-Image Generation

Figure 6: (a) Uncurated samples generated by class-conditioned image DART trained on ImageNet. (b) Selected samples generated by text-to-image DART trained on CC12M.

of the proposed improvement strategies. We also compare our methods with DiT (Peebles & Xie, 2022), using both 16 sampling steps (to match DART) and 250 steps (the suggested best setting). Notably, DART-AR achieves the best FID score of **3.98** among all variants and significantly surpasses DiT when using 16 steps, highlighting its advantage in leveraging generative trajectories, particularly when the number of sampling timesteps is limited. Not surprisingly, DiT with 250 steps performs better than DART. However, it is important to note that the official DiT model is trained for 7M steps, which is substantially more training iterations than those used for DART.

Figure 8 presents examples of generated results on ImageNet from all models using 16 sampling steps. DART-FM tends to produce sharper images with higher fidelity at higher CFG values. In contrast, DART-AR demonstrates an ability to generate more realistic samples at lower CFG values when compared to both the baselines and other variants.

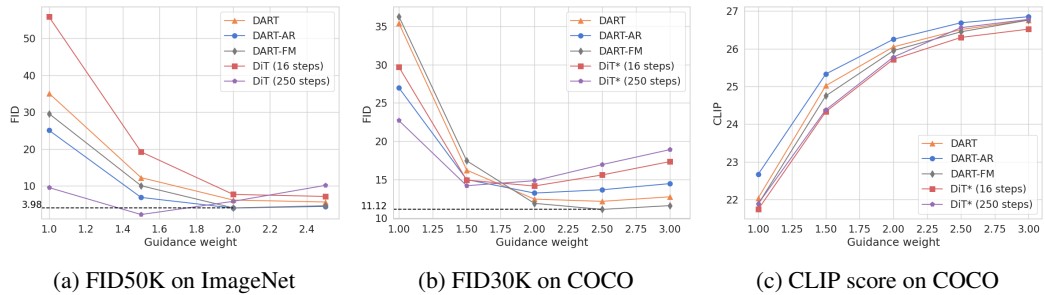

(a) FID50K on ImageNet      (b) FID30K on COCO      (c) CLIP score on COCO

Figure 7: Comparison of DART, DART-AR, DART-FM and baseline models with different CFG guidance scale on different benchmarks. * denotes models implemented and trained by us.

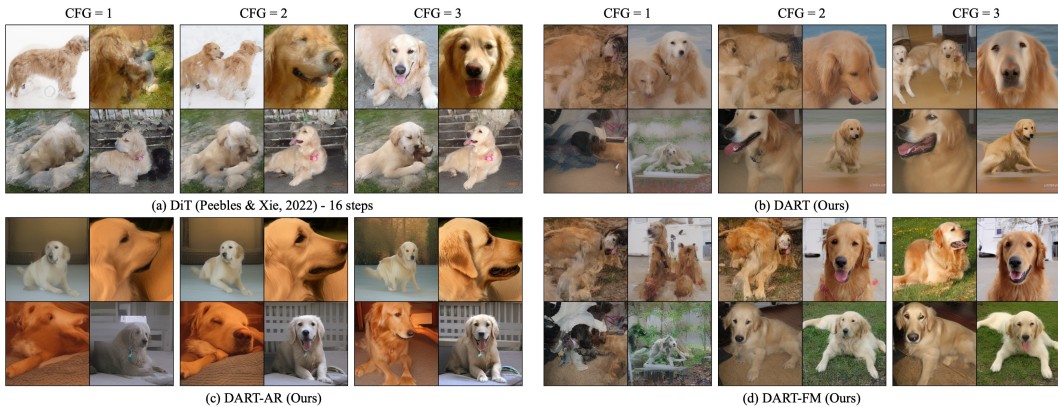

Figure 8: Sample comparison of DARTs and baseline DiT models with different CFG gudidance scale on ImageNet with the class label *golden retriever*.

**Text-to-Image Generation** To demonstrate the capability of DART at scale, we train the model for text-to-image generation. We also implement an in-house DiT with cross attention to text condition in comparison to our models, where we evaluate the performance on both 16 and 250 steps. As shown in Figure 7, while both -AR and -FM variants still show clear improvements against the default DART, FM achieves the best FID of **11.12**, indicating its ability of handling diverse generation tasks.

**Efficiency** We compare both the actual inference speed (measured by wall-clock time with batch size 32 on a single H100) as well as the theoretical computation (measured by GFlops) in 9(a). Since DART, DART-AR, DART-FM share the same encoder-decoder Transformer architecture, their flops are roughly the same. However, DART-AR has high wall clock inference time due to its large autoregressive steps, which have not been well parallelized in our current implementation. Integrated with recent advances in autoregressive LLMs, DART-AR can be deployed in a more efficient and we leave this for future investigations. DART-FM also have a inference time overhead due to the iterations of flow net in inference. Compared to DART, DiT has less flops for a single pass. However, it requires sufficient many number of iterations to generate high quality samples. DART has comparable flops and inference time as DiT with 16 sampling timesteps, while DART achieves better performance than DiT (16) on ImageNet and COCO, showing the efficiency benefits of DART.

**Scalibility** We show the scalability of our DART by training models of different sizes including small (S), base (B), large (L), and extra large (XL) on CC12M, where the configurations are listed in Appendix B.1. Figure 9(b) illustrates how the performance changes as model size increase. Across all the four models, CLIP score significantly improves by increasing the number of parameters in DART. Besides, the perform steadily increases as more training iterations are applied. This demonstrates that our proposed generative paradigm benefits from scaling as previous generative models like diffusion (Peebles & Xie, 2022) and autoregressive models (Li et al., 2024).

**Effects of noise levels.** We experiment with different noise levels, where we vary the number of total noise levels in our denoising framework (Figure 9(c)). A total of $T = 4$ or $8$ noise levels

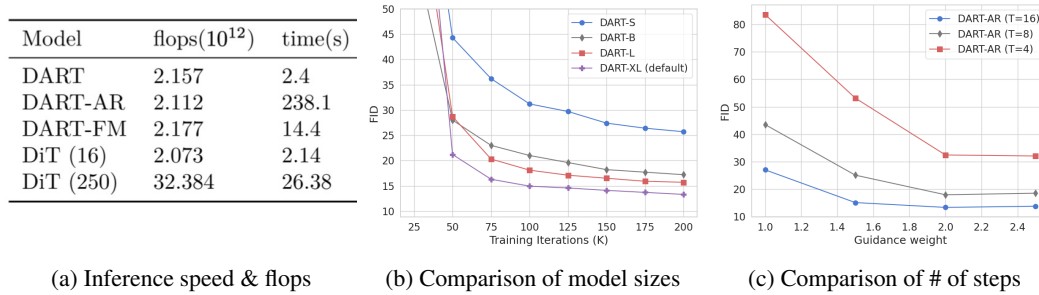

| Model | flops($10^{12}$) | time(s) |
|---|---|---|
| DART | 2.157 | 2.4 |
| DART-AR | 2.112 | 238.1 |
| DART-FM | 2.177 | 14.4 |
| DiT (16) | 2.073 | 2.14 |
| DiT (250) | 32.384 | 26.38 |

(a) Inference speed & flops     (b) Comparison of model sizes     (c) Comparison of # of steps

Figure 9: (a) Inference flops and wall clock time of different models. (b) Performance of DART of different sizes. (c) Effect of number of noise levels on DART.

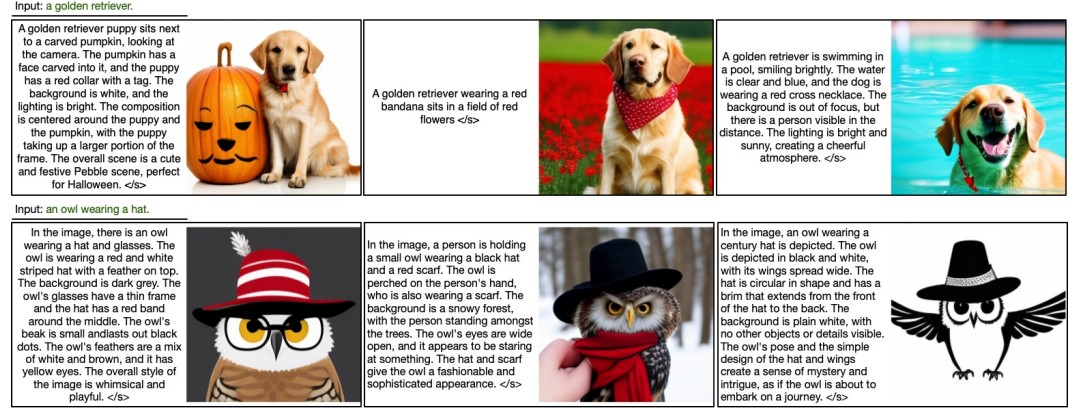

Figure 10: Examples of multi-modal generation with Kaleido-DART.

are trained in comparison with standard 16 noise levels. Not surprisingly, less noise levels lead to gradually degraded performance. However, with as few as 4 noise levels, DART can still generate plausible samples, indicating the capabilities of deploying DART in a more efficient way. Variants in noise levels provides a potential option for finding the optimal computation and performance trade-off especially when compute is limited.

**Multimodal Generation** We showcase the capabilities of our proposed method in the joint generation of discrete text and continuous images, as introduced in § 3.3. Figure 10 provides examples of multimodal generation using the Kaleido-DART framework. Given an input, the model generates rich descriptive texts along with corresponding realistic images, demonstrating its ability to produce diverse samples with intricate details. Notably, unlike Gu et al. (2024), our approach processes both text and images through the same model, utilizing a unified mechanism to handle both modalities. This unified framework can potentially be integrated into any multimodal language models.

## 5 CONCLUSION

We presented DART, a novel model that integrates autoregressive denoising with non-Markovian diffusion to improve the efficiency and scalability of image generation. By leveraging the full generation trajectory and incorporating token-level autoregression and flow matching, DART achieves competitive performance on class-conditioned and text-to-image tasks. This approach offers a unified and flexible solution for high-quality visual synthesis.

Our current model is restricted by the number of tokens in the denoising process. One direction is to explore architectures for long context modeling which enables application to problems like video generations. Also, current work only conduct preliminary investigates on multimodal generation. Future work may train a multimodal generative model based on the framework of the decoder-only language model to handle a wide variety of problems with one unified model.

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

# A  PROOF OF PROPOSITION 1

*Proof.* Due to $q(\boldsymbol{x}_t|\boldsymbol{x}_0) = \mathcal{N}(\boldsymbol{x}_t; \sqrt{\gamma_t}\boldsymbol{x}_0, (1-\gamma_t)\mathbf{I})$, we can write $\boldsymbol{x}_t = \sqrt{\gamma_t}\boldsymbol{x}_0 + \sqrt{1-\gamma_t}\boldsymbol{\epsilon}_t$.

With certain coefficient $\{\lambda_s^t\}_{s=t}^T$, let us define:

$$\boldsymbol{y}_t \triangleq \sum_{s=t}^{T} \lambda_s^t \boldsymbol{x}_s, \tag{11}$$

We study when the signal-to-noise ratio of $\boldsymbol{y}_t$ achieves its maximal value:

$$\boldsymbol{y}_t = \left( \sum_{s=t}^{T} \lambda_s^t \sqrt{\gamma_s} \right) \boldsymbol{x}_0 + \sum_{s=t}^{T} \lambda_s^t \sqrt{1-\gamma_s} \boldsymbol{\epsilon}_s \tag{12}$$

$$\Rightarrow \text{SNR}(\boldsymbol{y}_t) = \frac{\left( \sum_{s=t}^T \lambda_s^t \sqrt{\gamma_s} \right)^2}{\sum_{s=t}^T \lambda_s^{t\,2}(1-\gamma_s)} \leq \sum_{s=t}^{T} \frac{\gamma_s}{1-\gamma_s}, \tag{13}$$

which follows from Titu's lemma and the Cauchy-Schwarz inequality. It becomes an equality when:

$$\lambda_s^t \propto \frac{\sqrt{\gamma_s}}{1-\gamma_s}, \forall s \in [t, T] \tag{14}$$

Next, we demonstrate that $\{\boldsymbol{y}_t\}_t$ follows a Markov property when achieving the maximal signal-to-noise ratio for each $\boldsymbol{y}_t$. From Eq (14), let $\lambda_s^t = \rho_t \frac{\sqrt{\gamma_s}}{1-\gamma_s}, \rho_t > 0$, and $\eta_t = \frac{\gamma_t}{1-\gamma_t}, \bar{\eta}_t = \sum_{s=t}^T \eta_s$, we have:

$$\boldsymbol{y}_t = \rho_t \left( \bar{\eta}_t \boldsymbol{x}_0 + \sum_{s=t}^{T} \sqrt{\eta_s} \boldsymbol{\epsilon}_s \right) \tag{15}$$

$$= \rho_t \left( \bar{\eta}_t \boldsymbol{x}_0 + \sqrt{\bar{\eta}_t} \boldsymbol{\epsilon}_t' \right), \tag{16}$$

where we use $\boldsymbol{\epsilon}_t' \sim \mathcal{N}(0, \mathbf{I})$ to equivalently simplify the noise term. When $\rho_t = 1/\sqrt{\bar{\eta}_t^2 + \bar{\eta}_t}$, $\{\boldsymbol{y}_t\}_t$ is variance preserving (VP). Next, let us assume

$$\hat{\boldsymbol{y}} = \frac{\rho_{t+1}\bar{\eta}_{t+1}}{\rho_t\bar{\eta}_t} \boldsymbol{y}_t + \sigma\boldsymbol{\epsilon}, \boldsymbol{\epsilon} \sim \mathcal{N}(0, \mathbf{I}) \tag{17}$$

$$= \rho_{t+1}\bar{\eta}_{t+1}\boldsymbol{x}_0 + \rho_{t+1}\bar{\eta}_{t+1}/\sqrt{\bar{\eta}_t}\boldsymbol{\epsilon}_t' + \sigma\boldsymbol{\epsilon} \tag{18}$$

$$= \rho_{t+1} \left( \bar{\eta}_{t+1}\boldsymbol{x}_0 + \sqrt{\frac{\bar{\eta}_{t+1}^2}{\bar{\eta}_t} + \frac{\sigma^2}{\rho_{t+1}^2}} \boldsymbol{\epsilon}'' \right), \tag{19}$$

where we use $\boldsymbol{\epsilon}''$ to replace the noise term. So if we let $\hat{\boldsymbol{y}}$ match the distribution of $\boldsymbol{y}_{t+1}$, then

$$\frac{\bar{\eta}_{t+1}^2}{\bar{\eta}_t} + \frac{\sigma^2}{\rho_{t+1}^2} = \bar{\eta}_{t+1} \tag{20}$$

$$\Rightarrow \sigma^2 = \rho_{t+1}^2 \frac{\bar{\eta}_{t+1}\eta_t}{\bar{\eta}_t} > 0 \tag{21}$$

The above equation has root that $\sigma = \rho_{t+1}\sqrt{\bar{\eta}_{t+1}\eta_t/\bar{\eta}_t}$, This implies that we can find an independent noise term added to $\boldsymbol{y}_t$ to obtain $\boldsymbol{y}_{t+1}$, establishing that $\boldsymbol{y}_{tt}$ constitutes a *Markovian forward process*.

$$p(\boldsymbol{y}_{t+1}|\boldsymbol{y}_t) = \mathcal{N} \left( \frac{\rho_{t+1}\bar{\eta}_{t+1}}{\rho_t\bar{\eta}_t} \boldsymbol{y}_t, \rho_{t+1}^2 \frac{\bar{\eta}_{t+1}\eta_t}{\bar{\eta}_t} \mathbf{I} \right) \tag{22}$$

What's more, $\{\boldsymbol{x}_t\}_t$ sequence can be uniquely determined from $\{\boldsymbol{y}_t\}_t$ via

$$\boldsymbol{x}_t = \begin{cases} \left( \dfrac{\boldsymbol{y}_t}{\rho_t} - \dfrac{\boldsymbol{y}_{t+1}}{\rho_{t+1}} \right) \dfrac{1-\gamma_t}{\sqrt{\gamma}_t}, & \text{if } t < T \\[2ex] \dfrac{\boldsymbol{y}_t}{\rho_t} \dfrac{1-\gamma_t}{\sqrt{\gamma}_t}, & \text{if } t = T. \end{cases} \tag{23}$$

Therefore, the two processes $\{\boldsymbol{x}_t\}_t$ and $\{\boldsymbol{y}_t\}_t$ has a one-to-one correspondence.

$\square$

# B  IMPLEMENTATION DETAILS

## B.1  ARCHITECTURE

Table 1 lists the parameters of different model sizes for DART. For DART-FM, we implements the flow net as three feed-forward networks (FFNs) with additional adaptive LayerNorm for modulation. Also unlike common FFNs in Transformer, the hidden size stays unchanged in our implementation.

Table 1: Configurations of DART.

| Model | # Layers | Hidden dim | # Heads | # Params |
|---|---|---|---|---|
| DART-S | 12 | 384 | 6 | 48M |
| DART-B | 12 | 768 | 12 | 141M |
| DART-L | 24 | 1024 | 16 | 464M |
| DART-XL | 28 | 1280 | 20 | 812M |

## B.2  TRAINING

All models are trained with the following settings.

```
default training config:
    batch_size=128
    optimizer='AdamW'
    adam_beta1=0.9
    adam_beta2=0.95
    adam_eps=1e-8
    learning_rate=3e-4
    warmup_steps=10_000
    weight_decay=0.01
    gradient_clip_norm=2.0
    ema_decay=0.9999
    mixed_precision_training=bf16
```

# C  RESULTS ON IMAGENET-256

Table 2 lists the performance of our proposed DART in comparison to recent generative models on ImageNet-256. The reproduced results use the official codebase and checkpoint and we follow the best performing cfg scale reported in the original papers. We report reproduced results of DiT (Peebles & Xie, 2022), SiT (Ma et al., 2024), and MAR Li et al. (2024) with 16 sampling times which is the same as our vanilla DART. Our model achieves competitive performance when compared with diffusion models like LDM (Rombach et al., 2021) and AR models like VQGAN (Esser et al., 2021a) and RQ-Transformer (Lee et al., 2022). Admittedly, there is a gap between DART and SOTA visual generative models (like VAR (Tian et al., 2024) and MAR (Li et al., 2024)). However, we want to point out that many baselines are trained with significantly more FLOPs. For example, DiT (Peebles & Xie, 2022) is trained for 7M iterations whereas DART is only trained for 500k iterations. Also, baselines like VAR and MAR employ larger models than our DART. In particular, VAR deploys a 2B model while the largest DART model is approximately 800M. Besides, in our reproduced results, when using only 16 sampling steps as the setting of our vanilla DART, our model show significantly better performance than DiT, SiT and MAR. Also, we report MAR-AR (Li et al., 2024), a variant of MAR which generate tokens in an autoregressive manner instead of masked modeling which is applied in standard MAR models. DART which generates samples through autoregressive denoising shows better performance than MAR-AR. These results further validate the effectiveness of leveraging the whole denoising trajectory.

Table 2: Generative models on class-conditional ImageNet $256 \times 266$. *: reproduced from official codebase and checkpoints.

| Type | Model | FID↓ | IS↑ | #params | Steps |
|------|-------|------|-----|---------|-------|
| Diff. | ADM (Dhariwal & Nichol, 2021) | 10.94 | 101.0 | 554M | 250 |
| | CDM (Ho et al., 2022) | 4.88 | 158.7 | - | 8100 |
| | LDM (Rombach et al., 2021) | 3.60 | 247.7 | 400M | 250 |
| | DiT (Peebles & Xie, 2022) | 2.27 | 278.2 | 675M | 250 |
| | SiT (Ma et al., 2024) | 2.06 | 277.5 | 675M | 250 |
| AR | VQGAN (Esser et al., 2021b) | 15.78 | 74.3 | 1.4B | 256 |
| | RQTran (Lee et al., 2022) | 3.80 | 323.7 | 3.8B | 68 |
| | MAR-AR (Li et al., 2024) | 4.69 | 244.6 | 479M | 256 |
| | MAR (Li et al., 2024) | 1.55 | 303.7 | 943M | 256 |
| | VAR (Tian et al., 2024) | 1.73 | 350.2 | 2.0B | 10 |
| Reprod. | DiT* (Peebles & Xie, 2022) | 19.52 | 125.9 | 675M | 16 |
| | SiT* (Ma et al., 2024) | 6.98 | 122.9 | 675M | 16 |
| | MAR* (Li et al., 2024) | 6.37 | 221.3 | 943M | 16 |
| Ours | DART | 5.62 | 231.7 | 812M | 16 |
| | DART-AR | 3.98 | 256.8 | 812M | 4096 |
| | DART-FM | 3.82 | 263.8 | 820M | 16 |

## D ADDITIONAL TEXT-TO-IMAGE RESULTS

We here show more text-to-image generative examples from DART-AR and DART-FM at resolution $256 \times 256$ in Figure 13.

## E MULTI-RESOLUTION GENERATION

DART (including both the -AR and -FM variants) is a highly flexible framework that can be easily extended and applied in various scenarios with minimal changes in the formulation. For example, instead of learning a fixed resolution of images, one can learn a joint distribution of $p_\theta(\{x_0^i\}_{i=1}^N)$ where $x_0^i \in \mathbb{R}^{K_i \times C}$ is $x_0$ with a different resolution. Following the approaches proposed in (Gu et al., 2022; 2023; Zheng et al., 2023) for diffusion models, a single DART — referred to as *Matryoshka-DART* — can model multiple resolutions by representing each image with its corresponding noisy sequence $\{x_t^k\}_t$ separately, then flattening and concatenating these sequences for sequential prediction. As shown in Figure 11, we model the NOMAD objective in Eq (5) as

$$\max \mathcal{L}_\theta^{\text{Matryoshka}} = \mathbb{E}_{\{\{x_t^i\}_{t=0}^{T_i}\}_{i=1}^N \sim q(x_0)} \left[ \sum_{i=1}^N \sum_{t=1}^T \tilde{\omega}_t^i \cdot \log p_\theta(x_{t-1}^i | x_{t:T}^i, x_{1:T}^{<i}) \right], \qquad (24)$$

where the above can be implemented using any DART variant. Note that $x_0^i$ is not directly conditioned on $x_0^{<i}$, which not only avoids the need to handle shape changes at the boundaries but also mitigates potential error propagation, a common issue in learning cascaded diffusion (Ho et al., 2022). All low-resolution information is processed through self-attention.

By this approach, the model can balance the number of noise levels with the total number of tokens to achieve better efficiency. Additionally, learning resolutions can be progressively increased by finetuning low-resolution models with extended sequences.

Figure 12 visualizes the generative process of an image at resolution $256 \times 256$ and its upsampling to $512 \times 512$ by Matryoshka-DART. DART first iteratively refines the generative results for 16 denoising steps at resolution $256 \times 256$. It then upsamples images at resolution $512 \times 512$ and $1024 \times 1024$ through iterative denoising as well. Since the generation of high resolution is conditioned on the previous low-resolution samples, it needs less denoising steps at high resolution to generate realistic images. In particular, we add 4 denoising steps for generating $512 \times 512$ images and further add 2

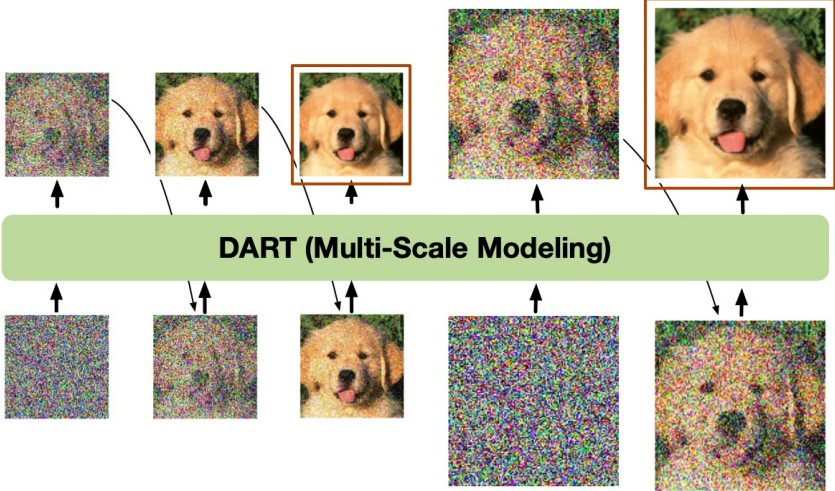

Figure 11: Illustrations of *Matryoshka-DART*. By joint training models, DART can perform multi-resolution.

Table 3: Comparison of DART variants.

| Model | Attn Mask | #AR steps | #FM steps |
|---|---|---|---|
| DART | Block-wise Causal | 16 | 0 |
| DART-AR | Causal | 4096 | 0 |
| DART-FM | Block-wise Causal | 16 | 1600 |

denoising steps for generating $1024 \times 1024$ images. Figures 14 and 15 show examples at resolution $512 \times 512$ and $1024 \times 1024$ from Matryoshka-DART finetuned from DART-FM at $256 \times 256$.

## F  COMPARISON OF DART VARIANTS

We here further clarify the differences and connections between the variants of DART. Table 3 lists the major comparison between DART, DART-AR, and DART-FM. Conceptually, DART predicts the denoised value and adds independent noise to acquire a less noisy image at each step. It conducts this denoising process autoregressively until the clean image is generated. DART-AR applies a token-wise autoregression instead of block-wise autoregressive in vanilla DART, which conduct denoising generation in a more fine-grained granularity. DART-FM, on the other hand, keeps the block-wise autoregression while introduces an additional flow network to conduct flow-mating-based refinement for generated tokens. Both DART-AR and DART-FM improve the performance over vanilla DART. In general, DART-FM demonstrates a better tradeoff between generation quality and inference efficiency.

Kaleido-DART is for multimodal text-image generation, which integrates next-token prediction for text and next-denoising prediction for images (proposed in our DART). In image generation, it can seamlessly adapt all three variants of proposed methods: DART, DART-AR, DART-FM. Similarly, Matryoshka-DART, which enables multi-resolution generation, also adapts to all the three DART variants. Since in Matryoshka-DART, one can simply concatenate high-resolution image tokens after the low-resolution ones, which doesn't affect the denoising modeling in these variants.

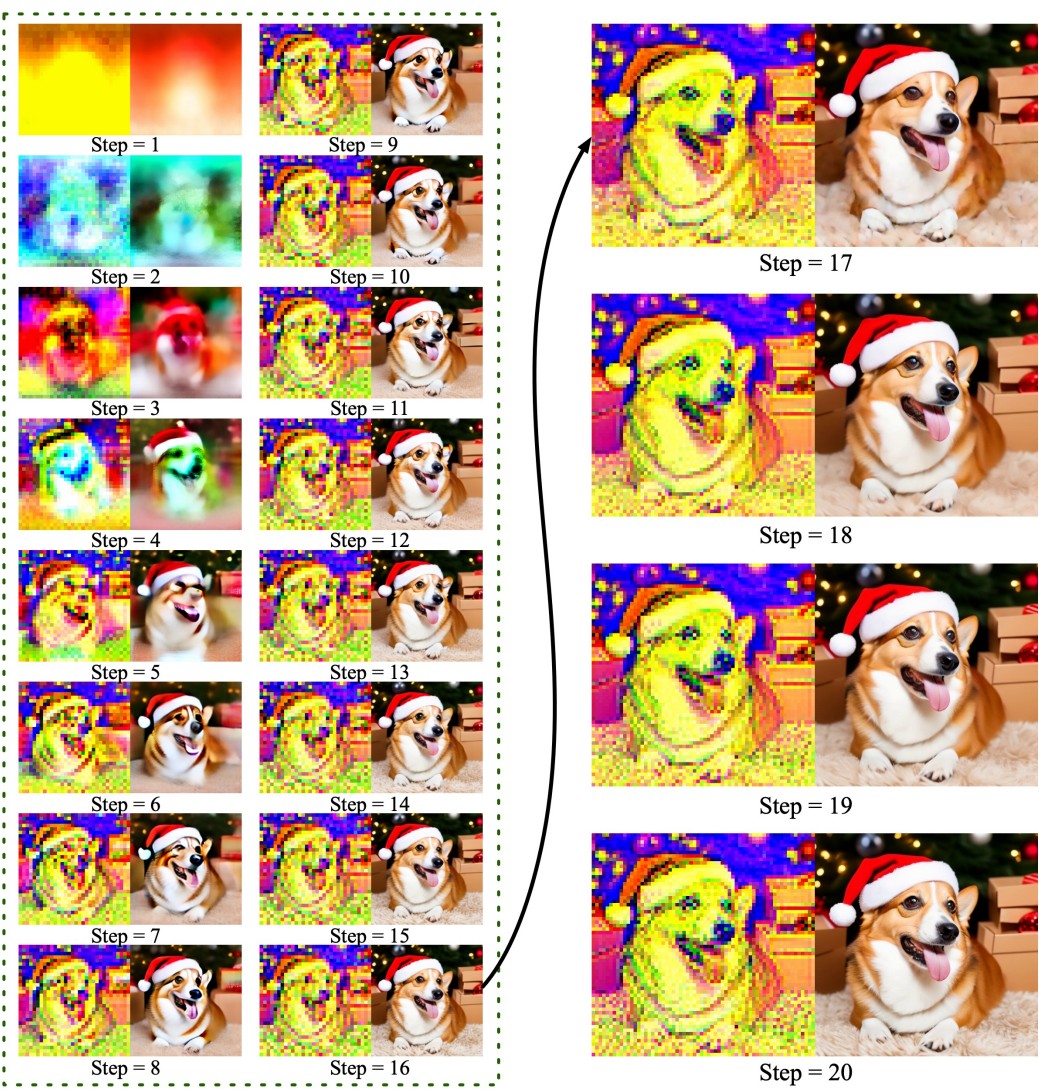

Figure 12: Visualization of the generation process (left: latent maps, right: decoded RGB images) for $256 \times 256$ ($T = 16$) and its upsampling to $512 \times 512$ ($T = 4$) using Matryoshka-DART.

(a) Generated text-to-image samples from DART-AR at 256x256 pixels

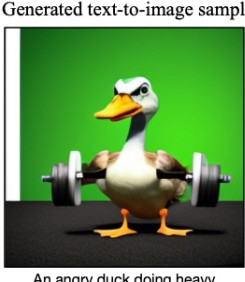 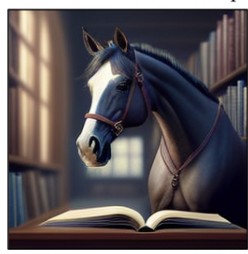 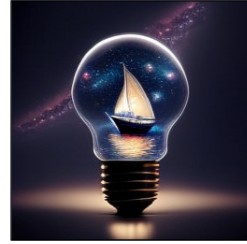

A corgi

An angry duck doing heavy weightlifting at the gym

A horse reading a book in a library

A light bulb containing a sailboat floats through the galaxy

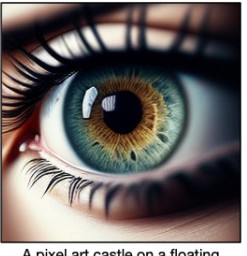 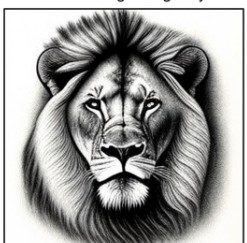

A pixel art castle on a floating island, with a waterfall of pixelated water cascading down into a pixelated cloud below.

An expressive oil painting of a chocolate chip cookie being dipped in a glass of milk, depicted as an explosion of flavors

A pixel art castle on a floating island, with a waterfall of pixelated water cascading down into a pixelated cloud below.

A pen drawing of a lion's face, with each strand of the mane meticulously detailed

(b) Generated text-to-image samples from DART-FM at 256x256 pixels

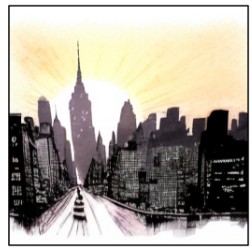 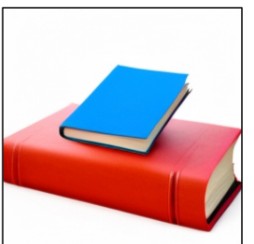 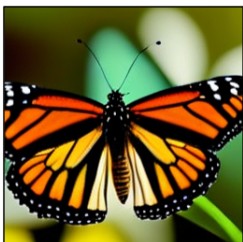 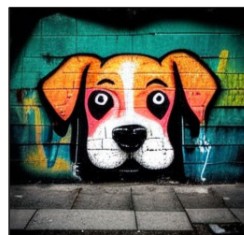

Downtown NYC at sunrise, detailed ink wash.

A small blue book sitting on a large red book.

A monarch butterfly.

graffiti of a funny dog on a street wall.

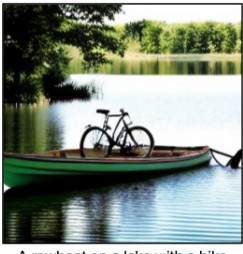 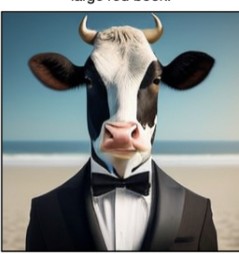 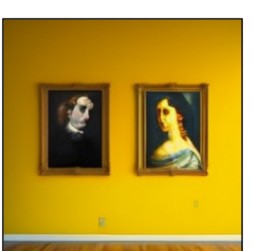 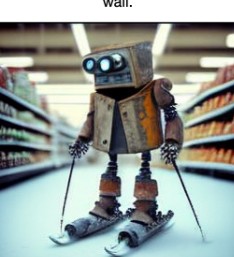

A rowboat on a lake with a bike on it.

A photo of a person with the head of a cow, wearing a tuxedo and black bowtie. Beach wallpaper in the background.

A yellow wall with two frames of famous oil paintings.

An old rusted robot wearing pants and a jacket riding skis in a supermarket.

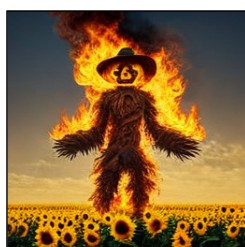 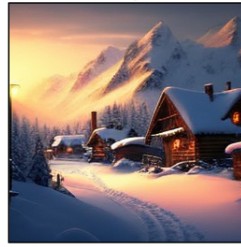 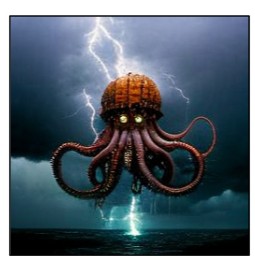 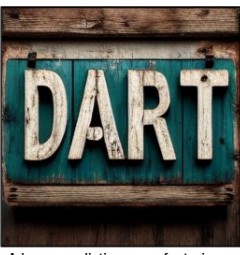

A giant scarecrow made entirely out of swirling fire, standing majestically in the landscape of sunflowers.

A snowy mountain village at dawn, with wooden cabins, and golden sunlight illuminating the snow-covered peaks

An enormous mechanical octopus emerging from the ocean, rising into a stormy sky filled with lightning.

A hyper-realistic scene featuring a weathered wooden sign prominently displaying the word 'DART' in bold.

Figure 13: Additional samples from DART varints on text-to-image generation at $256 \times 256$ pixels given various captions.

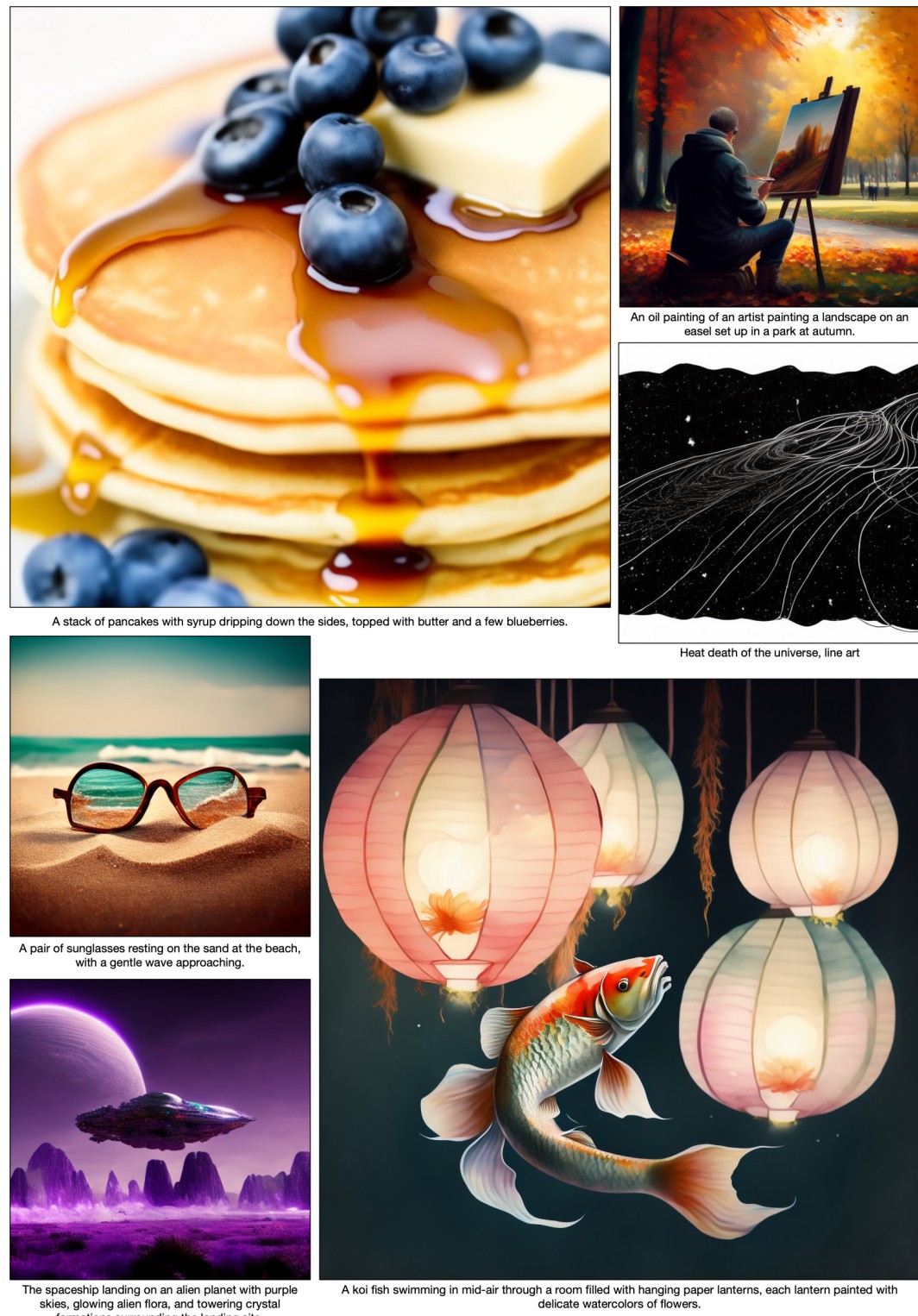

An oil painting of an artist painting a landscape on an easel set up in a park at autumn.

A stack of pancakes with syrup dripping down the sides, topped with butter and a few blueberries.

Heat death of the universe, line art

A pair of sunglasses resting on the sand at the beach, with a gentle wave approaching.

The spaceship landing on an alien planet with purple skies, glowing alien flora, and towering crystal formations surrounding the landing site.

A koi fish swimming in mid-air through a room filled with hanging paper lanterns, each lantern painted with delicate watercolors of flowers.

Figure 14: Samples from DART-FM with Matryoshka-DART fine-tuning on text-to-image generation at $512 \times 512$ and $1024 \times 1024$ pixels given various captions.

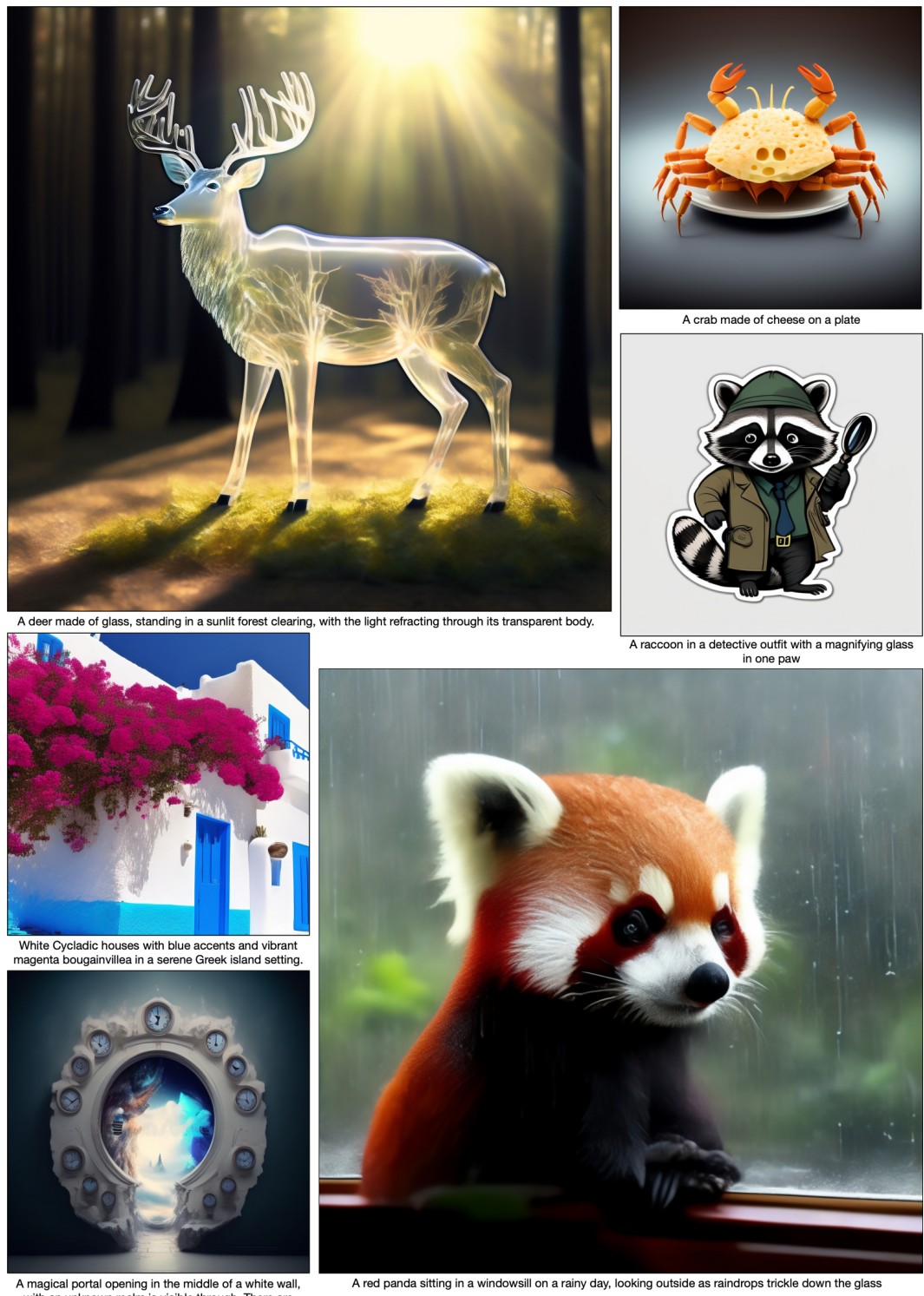

Figure 15: Samples from DART-FM with Matryoshka-DART fine-tuning on text-to-image generation at $512 \times 512$ and $1024 \times 1024$ pixels given various captions.

