# OpenReview forum: "Denoising Autoregressive Transformers for Scalable Text-to-Image Generation"
_ICLR.cc/2025/Conference — ICLR 2025 Poster_

### Official Review · Reviewer_Z9qZ · 2024-10-27

**Soundness:** 3
**Presentation:** 3
**Contribution:** 3
**Rating:** 5
**Confidence:** 4

**Summary:**

This paper presents DART, a model that integrates autoregressive and diffusion methods within a non-Markovian framework for efficient, scalable text-to-image generation. DART uses autoregressive denoising to retain the full generation trajectory, improving image coherence and generation quality. Key features include DART-AR for token-level autoregressive modeling and DART-FM for flow-based refinement, both enhancing image fidelity and flexibility. DART beats standard DiT in class-conditioned and text-to-image tasks and adapts well to multimodal generation, providing a unified and scalable approach for high-quality visual synthesis.

**Strengths:**

Overall I find that the writing is clear and well-structured, making it easy for readers to follow the arguments and understand the key points. This paper starts by revisiting key design choices of diffusion models and naturally denoising autoregressive transformer, which is a relatively novel framework aiming to integrate the strength of both diffusion and autoregressive models. Besides, this paper provides plenty of experiments and demos in terms of class/text-conditional image generation.

**Weaknesses:**

* My main concern is that while the paper’s title is Denoising Autoregressive Transformers for **Scalable** Text-to-Image Generation, I am worried about whether the proposed method can truly scale to higher-resolution images. In Line 264, it mentions using  T = 16 , which results in around 4000 tokens for a 256x256 image. So, for resolutions like 512 or 1024, can it really be trained and sampled efficiently using this framework? Although VAR also adopts a similar progressive generation approach, it uses different resolutions in its generation trajectory, whereas DART’s generation trajectory consists of full-sized, noisy images. I would like to see the experiments on higher resolutions (512 and 1024).
* The paper presents many different models (DART, DART-AR, DART-FM, Kaleido-DART), without providing a sufficient explanation of these models, such as their strengths and weaknesses. After reading this paper, I do not know how to choose between them since there is no consistent winner in the provided experiments. I would like to see a table to clearly describe differences, strengths, and weaknesses of each DART variant.
* The paper only compares the DART family with the original DiT on the ImageNet benchmark. However, DiT is originated from a paper two years ago, and there are a bunch of related works with improvements in both diffusion/autoregressive transformers that should be discussed and compared, e.g., SiT [1], VAR [2], MAR [3], etc... It is better to provide comparisons with these sota models in terms of FID scores and inference efficiency.

[1] Ma, Nanye, et al. "Sit: Exploring flow and diffusion-based generative models with scalable interpolant transformers." arXiv preprint arXiv:2401.08740 (2024).

[2] Tian, Keyu, et al. "Visual autoregressive modeling: Scalable image generation via next-scale prediction." arXiv preprint arXiv:2404.02905 (2024).

[3] Li, Tianhong, et al. "Autoregressive Image Generation without Vector Quantization." arXiv preprint arXiv:2406.11838 (2024).

**Questions:**

* Is the proposed DART compatible with existing techniques for diffusion models, e.g., DDIM, DPM-Solver, etc. I think it is a bit unfair to directly compare with original DiT with 16/256 steps without using any of these advanced samplers.

---

> ### Author Response · Authors · 2024-11-22
> **Official Response to Reviewer Z9qZ**
>
> 1. Q: My main concern is that whether the proposed method can truly scale to higher-resolution images. So, for resolutions like 512 or 1024, can it really be trained and sampled efficiently using this framework? Although VAR also adopts a similar progressive generation approach, it uses different resolutions in its generation trajectory, whereas DART’s generation trajectory consists of full-sized, noisy images. I would like to see the experiments on higher resolutions (512 and 1024).
>     - We thank the reviewers for bringing up this valuable question. **Our DART can scale to high-resolution images of resolution 512 and 1024 efficiently**. Instead of learning a fixed resolution of images, one can learn a joint distribution of $p_\theta(\{x^i_0\}_{i=1}^N)$ where $x^i_0\in \mathbb{R}^{K_i\times C}$ is $x_0$ with a different resolution. Following this formulation, a single DART can model multiple resolutions by representing each image with its corresponding noisy sequence $\{x^k_t\}_t$ separately, then flattening and concatenating these sequences for sequential prediction. Fig. 12 illustrates the denoising generative process of images with both resolutions $256 \times 256$ and $512 \times 512$.
>     - By this means, DART also benefits from progressive generation. Vanilla DART on resolution $256 \times 256$ uses 16 denoising steps. Since we follow this progressive generation paradigm, higher-resolution image generation is conditioned on generated $256 \times 256$ image, thus less denoising steps are needed, which greatly saves computation. Therefore, **DART may be even more scalable to higher-resolution generation than standard diffusion models, as DART can generate realistic images in very limited denoising steps.** In particular, we apply 4 denoising steps for image of $512 \times 512$ followed by 2 denoising steps for $1024 \times 1024$ image. We follow the default setting and set the patch size as 2 for $512 \times 512$ and $1024 \times 1024$ images, which ends up containing 1024 and 4096 tokens for each image separately. We finetuned a pretrained $256 \times 256$ DART on high resolution data. Fig. 14 and 15 show the text-to-image generation results of DART on resolutions of $512 \times 512$ and $1024 \times 1024$.
>
>
> 2. Q: The paper presents many different models (DART, DART-AR, DART-FM, Kaleido-DART), without providing a sufficient explanation of these models, such as their strengths and weaknesses. After reading this paper, I do not know how to choose between them since there is no consistent winner in the provided experiments. I would like to see a table to clearly describe differences, strengths, and weaknesses of each DART variant.
>     - We have included Appendix F in updated manuscript to further clarify the differences and connections between the variants of DART. Table below lists the major comparison between DART, DART-AR, and DART-FM. DART-AR applies a token-wise autoregression instead of block-wise autoregressive in vanilla DART, which conduct denoising generation in a more fine-grained granularity. DART-FM, on the other hand, keeps the block-wise autoregression while introduces an additional flow network to conduct flow-mating-based refinement for generated tokens. Both DART-AR and DART-FM improve the performance over vanilla DART. In general, DART-FM demonstrates a better tradeoff between generation quality and inference efficiency.
>         | Model | Attn Mask | #AR steps | #FM steps |
>         | -------- | ------- | -------- | -------- |
>         | DART        | Block-wise | 16       | 0 |
>         | DART-AR  | Causal        | 4096 | 0 |
>         | DART-FM | Block-wise | 16       | 1600 |
>     - Kaleido-DART is for multimodal text-image generation, which integrates next-token prediction for text and next-denoising prediction for images (proposed in our DART). In image generation, it can seamlessly adapt all three variants of proposed methods: DART, DART-AR, DART-FM. Similarly, Matryoshka-DART, which enables multi-resolution generation, also adapts to all the three DART variants. Since in Matryoshka-DART, one can simply concatenate high-resolution image tokens after the low-resolution ones, which doesn’t affect the denoising modeling in these variants.

---

> > ### Author Response · Authors · 2024-11-22
> > **Official Response to Reviewer Z9qZ (2)**
> >
> > 3. Q; The paper only compares the DART family with the original DiT on the ImageNet benchmark. There are a bunch of related that should be discussed and compared in terms of FID scores and inference efficiency.
> >     - We have added Table 2 to the updated manuscript to compare DART and recent baselines on class-conditioned image generation of ImageNet-256. Compared with diffusion models like LDM [1] and autoregressive (AR) models like VQGAN [2] and RQ-Transformer [3], DART achieves competitive performance. Admittedly, there is a gap between DART and SOTA visual generative models (like VAR [4] and MAR [5]). However, we want to point out that many baselines are trained with significantly more FLOPs. For example, DiT is trained for 7M iterations whereas DART is only trained for 500k iterations. Also, baselines like VAR and MAR employ larger models than our DART. In particular, VAR deploys a 2B model while the largest DART model is approximately 800M. Besides, in our reproduced results, when using only 16 sampling steps as the setting of our vanilla DART, our model show better performance than DiT [6], SiT [7], and MAR. Also, we report MAR-AR, a variant of MAR which generate tokens in an autoregressive manner instead of masked modeling which is applied in standard MAR models. DART which generates samples through autoregressive denoising shows better performance than MAR-AR. It further validates the effectiveness of leveraging the whole denoising trajectory. More details can be found in Response to All point 3 and Appendix C in updated manuscript.
> >
> > 4. Q: Is the proposed DART compatible with existing techniques for diffusion models, e.g., DDIM, DPM-Solver, etc. I think it is a bit unfair to directly compare with original DiT with 16/256 steps without using any of these advanced samplers.
> >    - Due to the non-Markovian formulation in DART which is different from standard diffusion models, our model does not allow easy applications to advanced sampling techniques like DPM-solver. We thank the reviewers for pointing out this valuable direction that would further improve DART.
> >    - In Table 2 of the updated manuscript, we also include comparison to reproduced SiT with 16 sampling steps using Dormand–Prince (RKDP) method as ODE solver. As shown, compared to models using advanced sampler, our proposed DART still shows competitive performance.
> >
> > References:
> >
> > [1] Rombach, Robin, Andreas Blattmann, Dominik Lorenz, Patrick Esser, and Björn Ommer. "High-resolution image synthesis with latent diffusion models." In Proceedings of the IEEE/CVF conference on computer vision and pattern recognition, pp. 10684-10695. 2022.
> >
> > [2] Esser, Patrick, Robin Rombach, and Bjorn Ommer. "Taming transformers for high-resolution image synthesis." In Proceedings of the IEEE/CVF conference on computer vision and pattern recognition, pp. 12873-12883. 2021.
> >
> > [3] Lee, Doyup, Chiheon Kim, Saehoon Kim, Minsu Cho, and Wook-Shin Han. "Autoregressive image generation using residual quantization." In Proceedings of the IEEE/CVF Conference on Computer Vision and Pattern Recognition, pp. 11523-11532. 2022.
> >
> > [4] Tian, Keyu, Yi Jiang, Zehuan Yuan, Bingyue Peng, and Liwei Wang. "Visual autoregressive modeling: Scalable image generation via next-scale prediction." arXiv preprint arXiv:2404.02905 (2024).
> >
> > [5] Li, Tianhong, Yonglong Tian, He Li, Mingyang Deng, and Kaiming He. "Autoregressive Image Generation without Vector Quantization." arXiv preprint arXiv:2406.11838 (2024).
> >
> > [6] Peebles, William, and Saining Xie. "Scalable diffusion models with transformers." In Proceedings of the IEEE/CVF International Conference on Computer Vision, pp. 4195-4205. 2023.
> >
> > [7] Ma, Nanye, Mark Goldstein, Michael S. Albergo, Nicholas M. Boffi, Eric Vanden-Eijnden, and Saining Xie. "Sit: Exploring flow and diffusion-based generative models with scalable interpolant transformers." arXiv preprint arXiv:2401.08740 (2024).

---

> > > ### Comment · Reviewer_Z9qZ · 2024-11-26
> > >
> > > Thank you for the detailed response. While the rebuttal addresses some of the questions raised, I believe the paper still has significant issues, particularly concerning the claim of **Scalable Generation**. My main concerns are as follows:
> > >
> > > * The proposed method for generating 512x512 and 1024x1024 images involves first generating a low-resolution image (e.g., 256x256) and then using very few denoising steps to generate higher-resolution images. This approach is similar to traditional cascaded generation and does not convincingly demonstrate that the DART framework itself can directly model higher resolutions. In fact, it appears that the model is essentially performing simple super-resolution.
> > > From the supplementary images, it is evident that the 512x512 and 1024x1024 images generated by DART exhibit noticeable visual artifacts and lack high-frequency details typically expected at higher resolutions. This suggests that the quality is not on par with using a dedicated super-resolution model, such as RealESRGAN, to upsample the images directly.
> > > Additionally, as shown in Figure 11, Matryoshka-DART requires all images—at different resolutions and noise levels—to be used simultaneously as context. This seemingly makes the framework more complex compared to traditional cascaded generation methods, rather than simplifying or improving the process.
> > >
> > > * In terms of model performance comparison, DART has not demonstrated metrics comparable to state-of-the-art diffusion or autoregressive methods. For example, the 800M DART model achieves an FID of approximately 4, which lags behind SOTA results. The authors attribute this to the shorter training time and smaller model size of DART. However, this explanation contradicts the paper's core claim of being a "scalable image generation model."
> > >
> > > * Additionally, the authors only report the number of sampling steps in the table, which is not a professional or sufficient comparison. Different models define "steps" differently, and this metric does not provide a direct basis for comparison. For instance, while DART-FM requires only 16 steps, it actually performs 1600 denoising computations. The paper should follow the practice of MAR and VAR and provide exact inference time or latency comparisons to give readers a more accurate understanding of the model's real-world efficiency and performance.
> > >
> > > * Furthermore, there are concerns regarding the reproduction of baseline results in Table 2. For example, according to Table 2 in the FlowTurbo [1] paper, SiT-XL achieves an FID of ~2 using simple solvers like dopri5 or heun's solver. However, the authors report an FID of ~7 for SiT, which is a significant discrepancy. This raises questions about the accuracy of the reported baseline results.
> > >
> > > * Finally, the authors admit that DART cannot directly adopt existing techniques from diffusion or flow-based models, such as DPM-Solver, nor do they compare against such methods. This further limits DART's potential as a scalable framework for the community to build upon.
> > >
> > > [1] Zhao, Wenliang, et al. "FlowTurbo: Towards Real-time Flow-Based Image Generation with Velocity Refiner." arXiv preprint arXiv:2409.18128 (2024).

---

> ### Author Response · Authors · 2024-11-26
> **Response to Followup Comments of Reviewer Z9qZ (part 1)**
>
> We thank the reviewer for detailed comments.
>
> We understand your major concern is on the capability of **scalable generation** of our proposed DART. We want to highlight that, from the perspective of this work, the scalability of a visual generative model contains two aspects:
> **(1) use compute in an efficient way especially as the model size grows, and (2) have the ability to easily generate images at any resolution.**
>
> For (1), DART is designed to leverage the whole denoising trajectory and the pre-computed hidden states, so that it can generate realistic results with less steps than standard diffusion model. Also, as shown in Figure 9b, DART benefits trivially from scaling up the model size, which demonstrates the scalability of proposed framework. For (2), we have further showcased the capability of DART in generating images at high-resolution of $512 \times 512$ and $1024 \times 1024$ by trivially connecting the denoising sequence of different resolutions without additional architectural/model change.
>
> **We respectfully notice that we may have discrepancy regarding the extensive terminology of *scalable generation*. And if this helps, we would be willing to change the title to “DART: Denoising Autoregressive Transformers for Visual Generation” to avoid the ambiguity of “scalable” in different meanings.**
> We believe this would not diminish our core contribution, our proposed method paves a way for a novel generative model with a more unified framework beyond diffusion and autoregressive models.
>
> Please find below point-by-point response to your comments.
>
> * *"It does not convincingly demonstrate that the DART framework itself can directly model higher resolutions. In fact, it appears that the model is essentially performing simple super-resolution..."*
>     * We want to highlight that our Matryoshka-DART for high-resolution generation is **different** from super resolution. Since the generation of high-resolution images are not dependent on the generated clean low-resolution image but the whole trajectory of noisy images. The low-resolution clean image is the output which is never fed back to DART. By this means, Matryoshka-DART naturally mitigates potential error propagation in conventional super-resolution models.
>     * Also, Matryoshka-DART models different resolutions by concatenating the tokens into one sequence, which doesn’t require extra efforts in handling shape change at boundaries as in traditional cascaded generation. Such a strategy further offers flexibility of modeling images of any ratios. We showcase results on squared images for simplicity purposes. But the model can be easily extended to other ratios by modeling the sequence of tokens. This model is also not restricted to model the same image of different resolutions.
>     * We also want to kindly point out that it’s a widely adapted paradigm for visual generation that the model is first trained on low-resolution (e.g., $256 \times 256$) and then tuned on high-resolution images (e.g., $512 \times 512$ and $1024 \times 1024$), like DALLE [1], SD3 [2], PixArt [3]. Here, we follow the mainstream of scalable generation to high-resolution. We adapt the pretraining on $256 \times 256$ and finetuning on higher-resolution strategy for training efficiency considerations.
>     * The model leverages its pretrained capability for low-resolution generation and the denoising trajectory of low-resolution images, enabling efficient training. However, it can also be trained from scratch for higher-resolution generation. Notably, after fine-tuning on higher-resolution data, DART retains its ability to generate low-resolution images, unlike conventional diffusion models, which are typically limited to high-resolution generation after fine-tuning.
>     * Finally, the high-resolution generation are not of low quality; in fact, they are relatively sharp compared to some existing works. While certain images may appear to lack fine details, this is primarily due to insufficient training during the rebuttal phase. In the next version, we will include a detailed comparison on ImageNet 512 to further demonstrate the model’s capabilities.
>
> * *"...This suggests that the quality of higher-resolution generation is not on par with using a dedicated super-resolution model, such as RealESRGAN, to upsample the images directly..."*
>     * We would like to clarify that our model, DART, is not a super-resolution model, as discussed earlier. While models like RealESRGAN are designed specifically for super-resolution tasks, generating high-resolution outputs from given low-resolution images, DART is fundamentally a generative model. Consequently, comparing DART to models like RealESRGAN may not be entirely appropriate. Additionally, DART is currently trained using only an L2 loss function. Incorporating GAN-based loss during training could further enhance its ability to produce sharper and more visually compelling results, which we plan to explore in future work.

---

> > ### Author Response · Authors · 2024-11-26
> > **Response to Followup Comments of Reviewer Z9qZ (part 2)**
> >
> > * *"Matryoshka-DART ... seemingly makes the framework more complex compared to traditional cascaded generation methods, rather than simplifying or improving the process."*
> >     * The way Matryoshka-DART models high-resolution images does NOT increase the framework’s complexity; on the contrary, it simplifies the generative process. Unlike conventional cascaded generative frameworks that require additional modules or training separate models, Matryoshka-DART generates images of different resolutions within a single model. It achieves this by concatenating *noisy image tokens* representing different resolutions into a unified sequence, enabling a streamlined and flexible framework for multi-resolution generation. This approach also extends the model’s flexibility, allowing it to effectively handle images of arbitrary aspect ratios, not just square images.
> >     * Second, DART leverages the entire denoising trajectory, requiring significantly fewer denoising steps to generate high-resolution images. Conventional super-resolution models rely solely on low-resolution inputs, whereas DART utilizes the full denoising trajectory of noisy low-resolution images. The low-resolution portion is end-to-end trained alongside the high-resolution outputs, effectively precomputing useful features that contribute to high-resolution generation in a more efficient manner (in this sense, similar to VAR [8]). As a result, when generating a $512 \times 512$ image, DART requires only 4 additional denoising steps for the high-resolution image, on top of 16 steps for the $256 \times 256$ image. This demonstrates that DART offers a simple and efficient framework for high-resolution and multi-resolution generation.
> >
> > * *"DART has not demonstrated metrics comparable to state-of-the-art diffusion or autoregressive methods ... However, this explanation contradicts the paper’s core claim of being a “scalable image generation model.”*
> >     * By “scalable generation,” we mean that DART leverages the entire denoising trajectory to produce realistic results with significantly less computational cost compared to standard diffusion models. Furthermore, as demonstrated in Figure 9b, DART exhibits only a marginal benefit from scaling up the model size, emphasizing its efficiency. We acknowledge that there may be differences in interpretation regarding the term “scalable generation.” If changing the title helps to address any potential misalignment, we are open to making such adjustments.
> >
> >
> > * *"The authors only report the number of sampling steps in the table, which is not a professional or sufficient comparison ... The paper should provide exact inference time or latency comparisons."*
> >     * We have included inference time comparison of DART and DiT [4] in **Figure 9a**. As shown, vanilla DART has almost the same inference time as DiT with 16 steps while DART achieves better performance.
> >     * We would also like to clarify that the reported number of steps does not include the diffusion steps in MAR. This is because both DART-FM and MAR involve a combination of autoregressive steps and diffusion/flow matching steps (the latter using only a lightweight 3-layer MLP), which have significantly different inference costs. Specifically, autoregressive steps are considerably more computationally expensive than diffusion/flow matching steps. Notably, the MAR results reported in their original paper include 25,600 diffusion steps.
> >
> > * *"Furthermore, there are concerns regarding the reproduction of baseline results in Table 2.*"
> >     * As mentioned in Appendix C, we use the official codebase and checkpoints from authors to reproduce DiT [4], SiT [5], and MAR [6]. In particular, we adapt default dopri5 solver with 16 sampling steps for SiT and report the results. We notice that Table 2 in FlowTurbo [7] applies 25 steps for SiT which has more steps than our setting. The difference of sampling steps can lead the discrepancy of the FID numbers.

---

> ### Author Response · Authors · 2024-11-26
> **Response to Followup Comments of Reviewer Z9qZ (part 3)**
>
> * *"Finally, the authors admit that DART cannot directly adopt existing techniques from diffusion or flow-based models, such as DPM-Solver, nor do they compare against such methods. This further limits DART’s potential as a scalable framework for the community to build upon."*
>     * We acknowledge that the current implementation of DART is not directly compatible with solvers developed for diffusion and flow-based models. However, as demonstrated in Proposition 1, there exists a **bijective mapping** between the non-Markovian diffusion process (DART) and the Markovian diffusion process (conventional diffusion). This mapping opens up the possibility of adapting diffusion solvers by transitioning the DART denoising process to a Markovian framework. Far from limiting DART’s potential for community-driven development, this adaptability presents opportunities to further enhance the framework, fostering a deeper unification of diffusion and autoregressive models.
>    * Unlike conventional diffusion models, DART features a consistent training and sampling denoising trajectory, enabling the optimization of the generation process in a single forward pass. Exploring the development of a learnable scheduler or integrating advanced solvers into the denoising trajectory presents valuable opportunities for the community to further enhance and build upon this framework.
>    * DART introduces a novel class of generative models that can be built on any non-Markovian process. In this paper, we focused on the basic setting of independent noise. Extending DART to incorporate more general non-Markovian processes represents a promising direction for future research, with the potential to significantly enhance model performance.
>
> References:
> [1] Ramesh, Aditya, Prafulla Dhariwal, Alex Nichol, Casey Chu, and Mark Chen. "Hierarchical text-conditional image generation with clip latents." arXiv preprint arXiv:2204.06125 1, no. 2 (2022): 3.
>
> [2] Esser, Patrick, Sumith Kulal, Andreas Blattmann, Rahim Entezari, Jonas Müller, Harry Saini, Yam Levi et al. "Scaling rectified flow transformers for high-resolution image synthesis." In Forty-first International Conference on Machine Learning. 2024.
>
> [3] Chen, Junsong, Jincheng Yu, Chongjian Ge, Lewei Yao, Enze Xie, Yue Wu, Zhongdao Wang et al. "Pixart-$\alpha $: Fast training of diffusion transformer for photorealistic text-to-image synthesis." arXiv preprint arXiv:2310.00426 (2023).
>
> [4] Peebles, William, and Saining Xie. "Scalable diffusion models with transformers." In Proceedings of the IEEE/CVF International Conference on Computer Vision, pp. 4195-4205. 2023.
>
> [5] Ma, Nanye, Mark Goldstein, Michael S. Albergo, Nicholas M. Boffi, Eric Vanden-Eijnden, and Saining Xie. "Sit: Exploring flow and diffusion-based generative models with scalable interpolant transformers." arXiv preprint arXiv:2401.08740 (2024).
>
> [6] Li, Tianhong, Yonglong Tian, He Li, Mingyang Deng, and Kaiming He. "Autoregressive Image Generation without Vector Quantization." arXiv preprint arXiv:2406.11838 (2024).
>
> [7] Zhao, Wenliang, Minglei Shi, Xumin Yu, Jie Zhou, and Jiwen Lu. "FlowTurbo: Towards Real-time Flow-Based Image Generation with Velocity Refiner." arXiv preprint arXiv:2409.18128 (2024).
>
> [8] Tian, Keyu, Yi Jiang, Zehuan Yuan, Bingyue Peng, and Liwei Wang. "Visual autoregressive modeling: Scalable image generation via next-scale prediction." NeurIPS 2024.

---

> > ### Comment · Reviewer_Z9qZ · 2024-11-29
> >
> > Thank you for your detailed response, which has addressed most of my concerns. I acknowledge that the proposed DART framework represents a novel generative model based on diffusion and autoregressive methods, and I appreciate its demonstrated potential for modeling images at various resolutions.
> >
> > However, my primary concern remains whether modeling the entire noise trajectory, as proposed, truly provides a meaningful advantage compared to existing generative frameworks such as diffusion, AR, MAR, VAR, and others. While the authors provide a comparison in Figure 9 against a naive DiT with 16/250 steps, it is evident that the better-performing variants of the proposed framework, namely DART-FM and DART-AR, require significantly more time to generate 256×256 images compared to both the baseline DART and DiT with 16 steps. Yet, the authors use DART-FM and DART-AR as examples to demonstrate the superior performance of the DART framework against DiT with 16 steps.
> >
> > For instance, existing methods like VAR and MAR can achieve much better results on ImageNet 256×256 within an inference time of ~0.2 seconds, which far surpasses the performance of DART. Therefore, I remain unconvinced that DART can be considered a competitive framework compared to other state-of-the-art generative models.

---

> > > ### Author Response · Authors · 2024-11-29
> > > **Response to the Additional Comment of Reviewer Z9qZ**
> > >
> > > We sincerely thank the reviewer for acknowledging that *our proposed DART framework represents a novel generative model and for recognizing its demonstrated potential in modeling images at various resolutions.*
> > > Your thoughtful feedback is greatly appreciated and has helped us clarify key aspects of our work.
> > >
> > > ---
> > >
> > > ## Addressing the Concerns About DART’s Competitiveness and Meaningful Advantages
> > >
> > > While we respect the concerns raised regarding DART’s competitiveness and its meaningful advantages compared to existing generative frameworks, we would like to address these points and provide further clarification:
> > >  1. **Vanilla DART vs. Existing Methods:**
> > >       We respectfully **disagree** with the statement that DART does not provide meaningful advantages.
> > >       * As shown in Figure 7 (and clarified as Figure 9 the reviewer mentioned) for both ImageNet and COCO, as well as the additional results presented in Table 2 of the rebuttal, the vanilla DART (**without -AR or -FM** improvements) consistently **outperforms** DiT and MAR under the same configurations (16 steps).
> > >       * Figure 7 further illustrates how different classifier-free guidance (CFG) settings affect results for different DART variants compared with the baseline DiT models.
> > >       * We believe, the key advantage stems from DART’s ability to utilize the context of the **entire noise trajectory**, which methods based on iterative refinement (e.g., DiT, MAR) cannot effectively leverage. These methods must recompute features at each step, constrained by the previous noisy input, while DART has no such "bottleneck".
> > > This demonstrates why DART is fundamentally different from traditional diffusion models and why we believe it represents an exciting direction for generative modeling.
> > >
> > >  2. **Use of DART-AR and DART-FM**
> > > We would like to clarify that we did not use DART-AR or DART-FM as sole examples to demonstrate superiority over DiT. Instead, these two improvements address specific *limitations* of the vanilla DART framework, which we honestly discussed in Section 3.2.
> > >      * The autoregessive approach introduces memory constraints, limiting its ability to model under independent Gaussian assumptions. DART-AR and DART-FM address this limitation by breaking these Gaussian assumptions, offering further improvements at the cost of additional inference time.
> > >      * As shown in Figure 7 and Table 2, these variants build on the vanilla DART formulation and provide clear enhancements.
> > >
> > >  3. **Comparison with MAR**
> > > While MAR is a strong baseline, as shown in our evaluations (e.g., Table 2), the results indicate that DART outperforms MAR under 16 step settings. Additionally, the inference time (~0.2s) reported for MAR is measured using a batch size of 256, which differs from the conditions of our speed measurements and is therefore not directly comparable.
> > >
> > >  4. **Comparison with VAR**
> > > We acknowledge that VAR currently outperforms DART in terms of quality and inference speed. However, we highlight several distinctions between the two methods:
> > >
> > >      * VAR operates in discrete space and requires training a multi-scale codebook, while DART is applied directly in **continuous** space without additional pretraining. This gives DART greater flexibility and eliminates information loss from discretization.
> > >      * VAR benefits from categorical softmax modeling in discrete space, which can be more expressive than DART’s Gaussian modeling. However, DART avoids the trade-off of discretization, preserving continuous information.
> > >      * **Both VAR and DART leverage generation history**—VAR through multi-scale generation, and DART through non-Markovian denoising trajectories. This shared benefit underscores the potential of DART while highlighting the flexibility of our approach.
> > >
> > > While VAR is currently stronger, we believe that DART has significant room for improvement, particularly in **addressing the limitations of Gaussian assumptions**. Exploring better ways to bridge the gap between denoising steps—e.g., relaxing Gaussian modeling or incorporating consistency models or GAN-based approaches—could further enhance DART without compromising inference speed.
> > >
> > > **Conclusion**
> > >
> > >
> > > We believe DART is a competitive generative framework, even at this early stage, with promising potential:
> > >  1. The vanilla DART outperforms existing baselines like DiT and MAR in similar configurations.
> > >  2. The extensions DART-AR and DART-FM show clear improvements and highlight pathways for addressing the limitations of the autoregressive formulation.
> > >  3. While not currently outperforming state-of-the-art methods like VAR, DART provides a novel and flexible framework for autoregressive denoising in continuous space, offering unique advantages and ample room for future development.
> > >
> > > ---
> > >
> > > We hope this clarification addresses the reviewer’s concerns and demonstrates the meaningful advantages of DART as a novel approach to generative modeling.

---

### Official Review · Reviewer_7f3i · 2024-11-02

**Soundness:** 4
**Presentation:** 3
**Contribution:** 3
**Rating:** 6
**Confidence:** 4

**Summary:**

DART unifies autoregressive (AR) and diffusion models within a non-Markovian framework, addressing the limitations of traditional diffusion models' Markovian property.

**Strengths:**

1. DART has well-grounded theoretical framework connecting non-Markovian and Markovian processes.

2. It can handle multiple modalities.

3. The experiments across different tasks are comprehensive.

**Weaknesses:**

1. While the technical contributions are solid, the paper's narrative structure prioritizes theoretical formulation over an intuitive explanation of the approach. The image is first transformed into several patches in the latent space through a Variational Autoencoder (VAE). The basic version of DART processes these patches simultaneously, predicting their denoised mean values, and then continues denoising in a manner similar to traditional diffusion models. The authors can include a more accessible explanation of how DART works in practice.

2. Even though, I think this paper's results are good, and show generalization ability, I am not sure whether this AR+diffusion method become new trend for future visual generation. Is is better than transfusion, show-o, and mar manner? (I recognize those should be concurrent work. This is just for curiosity, and I will not penalize the paper in terms of scoring).

**Questions:**

see Weaknesses.

---

> ### Author Response · Authors · 2024-11-22
> **Official Response to Reviewer 7f3i**
>
> 1. Q: The paper's narrative structure prioritizes theoretical formulation over an intuitive explanation of the approach. The image is first transformed into several patches in the latent space through a Variational Autoencoder (VAE). The basic version of DART processes these patches simultaneously, predicting their denoised mean values, and then continues denoising in a manner similar to traditional diffusion models. The authors can include a more accessible explanation of how DART works in practice.
>     - We have added the following descriptions to Appendix F. “Conceptually, DART predicts the denoised value and adds independent noise to acquire a less noisy image at each step. It conducts this denoising process autoregressively until the clean image is generated.”
>
>
> 2. Q: I am not sure whether this AR+diffusion method become new trend for future visual generation. Is it better than transfusion, show-o, and mar manner? (I recognize those should be concurrent work. This is just for curiosity, and I will not penalize the paper in terms of scoring).
>     - We thank the reviewer for bringing up this valuable discussion. There are definitely works in attempt to integrating autoregressive (AR) and diffusion models. Transfusion and Show-O develop multimodal vision-language models (VLMs) but they apply AR for text while applying diffusion for image generation separately. Such that they integrate AR and diffusion models in a brutal way. Namely different losses are used to train the two aspects. In DART, image generation is also modeled as an autoregressive denoising which can be integrated seamlessly with AR text generation as shown in preliminary results of Kaleido-DART.
>     - On the other hand, MAR autoregressively predicts a latent for each patch and applies an additional diffusion model to generate the image tokens conditioned on the predicted latent. DART integrates the denoising process into the autoregressive modeling and doesn’t require extra diffusion module. By this means, DART enables a more unified framework that combines AR and diffusion models.
>     - It is still uncertain which model will be the best way to integrate AR and diffusion models. As this area is still a growing rapidly. However, the results shown in the paper already demonstrate the capabilities of DART that conventional diffusion or AR model cannot achieve trivially. We believe DART paves a way for developing novel generative models beyond AR and diffusion models and may lead to a more unified framework. And we would like to further improve the framework in future works.

---

> > ### Comment · Reviewer_7f3i · 2024-11-29
> >
> > My questions are addressed, and thus I keep my score as 6

---

> > > ### Author Response · Authors · 2024-11-29
> > >
> > > We thank the reviewer for recognizing our explanation. Please let us know if there are any additional questions we can further address.

---

### Official Review · Reviewer_AfU4 · 2024-11-02

**Soundness:** 3
**Presentation:** 3
**Contribution:** 3
**Rating:** 6
**Confidence:** 3

**Summary:**

The paper, introduces DART, a novel generative model that integrates autoregressive and diffusion approaches within a non-Markovian framework. Key contributions of the work include:
- DART Model: A non-Markovian diffusion model that uses autoregressive techniques to capture the full denoising trajectory, improving generation efficiency and flexibility.
- DART-AR and DART-FM Modules: DART-AR enables token-level autoregressive modeling for enhanced control and quality, while DART-FM employs flow-based refinement to increase model expressiveness.

**Strengths:**

1. The integration of autoregressive and non-Markovian diffusion in a unified framework is innovative, particularly by moving beyond the Markovian assumptions that traditionally constrain diffusion models. DART's AR and FM extensions add novel approaches for token-based and flow-matching refinement in image generation.
2. Experiments are carried out on low-resolution class-conditional and text-to-image generation as well as multimodal generation, demonstrating its effectiveness on different classes.

**Weaknesses:**

1. My biggest concern is the lack of comparison to SOTA approach, either in the form of diffusion models or auto-regressive models, for image generation tasks. The whole paper literally only shows comparison to the baseline DiT model which is published in 2022. There're a flourished set of literature for transformer-based image generation (SD-3[1], PixArt[2], etc.) and multimodal generation (Chameleon[3], Transfusion[4], etc.), yet none of these is compared in the paper. As such, it is really hard to understand the performance and actual benefits this approach brings us.
2. The efficiency of the model is a big concern. As shown in Fig 7 and Fig. 9(a), although a finer grain modeling by DART-AR could bring a marginal improvement, it would introduce 100x more latency, making it hard to justify whether this efficiency-performance tradeoff is ideal. Scaling up a diffusion model that consume 100x more latency may have better performance. This comparison would be good to show in Fig 7.

[1] "Scaling rectified flow transformers for high-resolution image synthesis." ICML 2024.
[2] "Fast Training of Diffusion Transformer for Photorealistic Text-to-Image Synthesis". ICLR2024
[3] "Chameleon: Mixed-Modal Early-Fusion Foundation Models". ArXiv
[4] "Transfusion: Predict the Next Token and Diffuse Images with One Multi-Modal Model". ArXiv

**Questions:**

Please refer to the weakness section.

---

> ### Author Response · Authors · 2024-11-22
> **Official Response to Reviewer AfU4**
>
> 1. Q: Lack of comparison to SOTA approach, either in the form of diffusion models or auto-regressive models, for image generation tasks.
>     - We have added Table 2 to the updated manuscript to include recent baselines of diffusion and autoregressive models on class-conditioned image generation of ImageNet-256. Compared with diffusion models like LDM [1] and autoregressive (AR) models like VQGAN [2] and RQ-Transformer [3], DART achieves competitive performance. Admittedly, there is a gap between DART and SOTA visual generative models (like VAR [4] and MAR [5]). However, we want to point out that many baselines are trained with significantly more FLOPs. For example, DiT is trained for 7M iterations whereas DART is only trained for 500k iterations. Also, baselines like VAR and MAR employ larger models than our DART. In particular, VAR deploys a 2B model while the largest DART model is approximately 800M. Besides, in our reproduced results, when using only 16 sampling steps as the setting of our vanilla DART, our model show better performance than DiT [6], SiT [7] and MAR. Also, we report MAR-AR, a variant of MAR which generate tokens in an autoregressive manner instead of masked modeling which is applied in standard MAR models. DART which generates samples through autoregressive denoising shows better performance than MAR-AR. It further validates the effectiveness of leveraging the whole denoising trajectory. More details can be found in Response to All point 3 and Appendix C in updated manuscript.
>
> 2. Q: The efficiency of the model is a big concern. As shown in Fig 7 and Fig. 9(a), although a finer grain modeling by DART-AR could bring a marginal improvement, it would introduce 100x more latency, making it hard to justify whether this efficiency-performance tradeoff is ideal.
>     - We agree with the reviewer that it’s valuable to explore the tradeoff between performance improvement and the sampling efficiency. One way is to develop an adaptive sampling strategy where more tokens are output at once in the beginning while less tokens are output later. Since we may want to allocate more FLOPs to better refine the generative results that are closer to clean data. Also, efficient architecture like state-space models may increase the efficiency over Transformer in modeling the denoising process. We believe these are valuable directions to explore as future works.
>
> References:
>
> [1] Rombach, Robin, Andreas Blattmann, Dominik Lorenz, Patrick Esser, and Björn Ommer. "High-resolution image synthesis with latent diffusion models." In Proceedings of the IEEE/CVF conference on computer vision and pattern recognition, pp. 10684-10695. 2022.
>
> [2] Esser, Patrick, Robin Rombach, and Bjorn Ommer. "Taming transformers for high-resolution image synthesis." In Proceedings of the IEEE/CVF conference on computer vision and pattern recognition, pp. 12873-12883. 2021.
>
> [3] Lee, Doyup, Chiheon Kim, Saehoon Kim, Minsu Cho, and Wook-Shin Han. "Autoregressive image generation using residual quantization." In Proceedings of the IEEE/CVF Conference on Computer Vision and Pattern Recognition, pp. 11523-11532. 2022.
>
> [4] Tian, Keyu, Yi Jiang, Zehuan Yuan, Bingyue Peng, and Liwei Wang. "Visual autoregressive modeling: Scalable image generation via next-scale prediction." arXiv preprint arXiv:2404.02905 (2024).
>
> [5] Li, Tianhong, Yonglong Tian, He Li, Mingyang Deng, and Kaiming He. "Autoregressive Image Generation without Vector Quantization." arXiv preprint arXiv:2406.11838 (2024).
>
> [6] Peebles, William, and Saining Xie. "Scalable diffusion models with transformers." In Proceedings of the IEEE/CVF International Conference on Computer Vision, pp. 4195-4205. 2023.
>
> [7] Ma, Nanye, Mark Goldstein, Michael S. Albergo, Nicholas M. Boffi, Eric Vanden-Eijnden, and Saining Xie. "Sit: Exploring flow and diffusion-based generative models with scalable interpolant transformers." arXiv preprint arXiv:2401.08740 (2024).

---

> > ### Comment · Reviewer_AfU4 · 2024-11-29
> >
> > Thanks for the response from the reviewer, I'll remain my rating.

---

### Official Review · Reviewer_xQrW · 2024-11-02

**Soundness:** 4
**Presentation:** 3
**Contribution:** 4
**Rating:** 8
**Confidence:** 4

**Summary:**

The paper presents DART, al generative model combining diffusion and autoregressive (AR) frameworks for efficient and scalable image synthesis. DART addresses limitations in traditional diffusion models, particularly their Markovian process, which restricts generative efficiency by only leveraging information from the previous denoising step.

**Strengths:**

The idea is interesting, exploring if we can use AR to model the image diffusion and denoising processes to generate image.

1. The technique is well-supported by a theory.

2. The results are good on border benchmarks. The T2I results seem promising.

**Weaknesses:**

My major concern is that the comparison is not comprehensive. DART integrates diffusion in the AR model, using the AR to model the diffusion. MAR [1] uses diffusion for the AR model, using the diffusion to model the tokenization. Even though their two different approaches, the comparison should include MAR, as MAR has already shown good generalization ability on multimodal generation [2]. It is becoming a trend. With this comparison, we can see if DART has this potential.

[1] MAR, NeurIPS 2024

[2] Fluid: Scaling Autoregressive Text-to-image Generative Models with Continuous Tokens

**Questions:**

1. In the inference stage, does the DART-FM also show the short-step sampling property? How to realize short-step sampling with DART-FM.

2. I still have no idea why DART-AR is better than DART. For me, the whole image processing mechanism should work better. Could the authors give me more detailed explanation?

---

> ### Author Response · Authors · 2024-11-22
> **Official Response to Reviewer xQrW**
>
> 1. Q: The comparison is not comprehensive. DART integrates diffusion in the AR model, using the AR to model the diffusion. MAR uses diffusion for the AR model, using the diffusion to model the tokenization. Even though their two different approaches, the comparison should include MAR, as MAR has already shown good generalization ability on multimodal generation. It is becoming a trend. With this comparison, we can see if DART has this potential.
>     - DART is connected with MAR as both methods integrate diffusion and autoregressive models. MAR autoregressively predicts a latent for each patch and applies an additional diffusion model to generate the image tokens conditioned on the predicted latent. Whereas our DART integrates the denoising process into the autoregressive modeling and doesn’t require extra diffusion module. Also, the best performing setting of MAR resembles masked modeling more instead of conventional autoregressive (AR) modeling. As shown in the MAR paper, with conventional AR, the performance greatly falls behind masked modeling setting.
>     - We agree that it would be beneficial to include compare DART with MAR. We have added comparison to MAR and other baselines in Table 2 of the updated manuscript. Admittedly, MAR achieves better performance than DART. However, MAR uses more sampling steps than vanilla DART. We also include comparison to reproduced MAR with 16 autoregressive steps in inference which follows a more similar setting as DART. As shown, with limited inference steps, DART achieves competitive performance with FID 3.98 vs MAR-16 with FID 6.37. This demonstrates the effectiveness of DART in leveraging the whole denoising trajectory.
>
> 2. Q: In the inference stage, does the DART-FM also show the short-step sampling property? How to realize short-step sampling with DART-FM.
>     - If we understand correctly, “short-step sampling property” refers to DART can generate realistic images in limited number of denoising steps. DART-FM uses the same number of autoregressive steps as vanilla DART (i.e., 16 steps). Additionally, DART-FM includes a small flow-matching module which refines the prediction at each denoising step. We implement a small flow network which only adds about 1% of total parameters. To further increase the efficiency of DART-FM, one can further decrease the sampling steps for flow-matching module. One strategy could be adaptive module where more sampling budget is applied when closer to clean data. Also, distillation methods can help significantly reduce the sampling steps. We believe these are valuable directions to explore as future works.
>
> 3. Q: I still have no idea why DART-AR is better than DART. For me, the whole image processing mechanism should work better. Could the authors give me more detailed explanation?
>     - Both DART and DART-AR have access to all the previous image patches through the denoising process. Therefore, both strategies enjoy the benefit of “whole image processing”. We attribute the better performance of DART-AR over DART to the more refined generative granularity. DART outputs a whole image at once which leaves no room for the model to refine the prediction of individual patches in the image. On the other hand, DART-AR outputs one patch as a time which allows the model to correct the imperfect prediction of previous patches. DART-AR conducts denoising in a more refined granularity than DART (patch v.s. image) which leads to better generative results.

---

> > ### Comment · Reviewer_xQrW · 2024-11-27
> > **Raise my score to 8**
> >
> > The authors solved my concerns, and I decide to raise my score to 8.

---

> > > ### Author Response · Authors · 2024-11-29
> > >
> > > We thank the reviewer for recognizing our explanation.

---

### Official Review · Reviewer_74aA · 2024-11-03

**Soundness:** 3
**Presentation:** 3
**Contribution:** 3
**Rating:** 6
**Confidence:** 3

**Summary:**

This paper proposes DART, a new method to learn a distribution over images by autoregressively denoising a noise image to obtain high-quality generated images. They claim that DART leverages all the latents within the trajectory in oppose to existing methods that only use the previous latent. They compare their method with DiT and observe improvements in generated contents as evaluated by FID and CLIP Score.

**Strengths:**

The method is scalable and efficient as explained in the paper.

**Weaknesses:**

I think that the comparison to existing work is not comprehensive. Is DiT the only other model that this work should be compared with?

**Questions:**

I want the authors to clarify the comprehension of their evaluation. Also, I'd want to see how the autoregressive generation is the cause of improvements. Can authors run any experiments supporting the fact that autoregressive generation, where attention happens with all previous tokens, has a huge role?

---

> ### Author Response · Authors · 2024-11-22
> **Official Response to Reviewer 74aA**
>
> 1. Q: Comparison to existing work is not comprehensive
>    - We have added Table 2 to the updated manuscript to compare DART and recent baselines on class-conditioned image generation of ImageNet-256. Compared with diffusion models like LDM [1] and autoregressive (AR) models like VQGAN [2] and RQ-Transformer [3], DART achieves competitive performance. Admittedly, there is a gap between DART and SOTA visual generative models (like VAR [4] and MAR [5]). However, we want to point out that many baselines are trained with significantly more FLOPs. For example, DiT is trained for 7M iterations whereas DART is only trained for 500k iterations. Also, baselines like VAR and MAR employ larger models than our DART. In particular, VAR deploys a 2B model while the largest DART model is approximately 800M. Besides, in our reproduced results, when using only 16 sampling steps as the setting of our vanilla DART, our model show better performance than DiT [6], SiT [7] and MAR. Also, we report MAR-AR, a variant of MAR which generate tokens in an autoregressive manner instead of masked modeling which is applied in standard MAR models. DART which generates samples through autoregressive denoising shows better performance than MAR-AR. More details can be found in Response to All point 3 and Appendix C in updated manuscript.
>
> 2. Q: How the autoregressive generation is the cause of improvements. Can authors run any experiments supporting the fact that autoregressive generation, where attention happens with all previous tokens, has a huge role?
>     - We thank the reviewer for bringing up this interesting discussion. In fact, DART won’t work if only tokens of one previous denoising step are given as standard diffusion model. This is because DART follows a non-Markovian formulation where the current state $x_t$ relies on all the previous states $x_{t+1:T}$. In practice, at each denoising step, individual noise is re-sampled to corrupt the data following the noise schedule. Whereas in standard Markovian diffusion process, the previous state $x_{t+1}$ alone is assumed to contain all the information needed to predict next state. In our early experiments, we found that if whole previous trajectory is not provided, the model failed to generate meaningful results. Such design is one of the major novelties in proposed DART which allows leverage of whole denoising trajectory.
>
> References:
>
> [1] Rombach, Robin, Andreas Blattmann, Dominik Lorenz, Patrick Esser, and Björn Ommer. "High-resolution image synthesis with latent diffusion models." In Proceedings of the IEEE/CVF conference on computer vision and pattern recognition, pp. 10684-10695. 2022.
>
> [2] Esser, Patrick, Robin Rombach, and Bjorn Ommer. "Taming transformers for high-resolution image synthesis." In Proceedings of the IEEE/CVF conference on computer vision and pattern recognition, pp. 12873-12883. 2021.
>
> [3] Lee, Doyup, Chiheon Kim, Saehoon Kim, Minsu Cho, and Wook-Shin Han. "Autoregressive image generation using residual quantization." In Proceedings of the IEEE/CVF Conference on Computer Vision and Pattern Recognition, pp. 11523-11532. 2022.
>
> [4] Tian, Keyu, Yi Jiang, Zehuan Yuan, Bingyue Peng, and Liwei Wang. "Visual autoregressive modeling: Scalable image generation via next-scale prediction." arXiv preprint arXiv:2404.02905 (2024).
>
> [5] Li, Tianhong, Yonglong Tian, He Li, Mingyang Deng, and Kaiming He. "Autoregressive Image Generation without Vector Quantization." arXiv preprint arXiv:2406.11838 (2024).
>
> [6] Peebles, William, and Saining Xie. "Scalable diffusion models with transformers." In Proceedings of the IEEE/CVF International Conference on Computer Vision, pp. 4195-4205. 2023.
>
> [7] Ma, Nanye, Mark Goldstein, Michael S. Albergo, Nicholas M. Boffi, Eric Vanden-Eijnden, and Saining Xie. "Sit: Exploring flow and diffusion-based generative models with scalable interpolant transformers." arXiv preprint arXiv:2401.08740 (2024).

---

> > ### Comment · Reviewer_74aA · 2024-11-25
> >
> > Thanks for the rebuttal and the explanation, the draft looks better now. I maintain my rating.

---

> > > ### Author Response · Authors · 2024-11-25
> > > **Official Response to Reviewer 74aA (2)**
> > >
> > > We thank the reviewer for the timely reply and helping improve the draft. Please let us know if there are any additional questions we can further address.

---

### Author Response · Authors · 2024-11-22
**Official Response to All Reviewers**

We thank all reviewers for their comments that help substantially improve the quality of our manuscript. We have updated the manuscript with changes highlighted. We here list the major updates and clarifications in response to the reviews.

1. First of all, we want to highlight the scope of this work is to build a **novel and potentially more scalable generative framework beyond existing popular autoregressive (AR) models and diffusion models**. In light of this, we propose DART, a Transformer-based generative model that implements non-Markovian diffusion process. DART generates images through iterative refinement like diffusion models. However, DART differs from diffusion model in the way that it allows access to the full denoising trajectory rather than a Markovian process which allows access to one previous state. Also, unlike AR models which usually relies on discrete tokens, our DART directly operate on continuous latents. DART adapts the iterative refinement through denoising which conventional AR lacks. DART showcases its capabilities in class-conditioned image generation, text-to-image generation, multi-resolution generation, and multimodal co-generation. We believe this work paves a way for developing novel generative models beyond AR and diffusion models and may lead to a more unified framework.

2. We want to further emphasize that our proposed model is different from several recent works on visual generation with autoregressive models.
    - MAR [1] is an AR model on continuous image tokens. It autoregressively predicts a latent for each patch and applies an additional diffusion model to generate the image tokens conditioned on the predicted latent. It differs from DART from two aspects. (1) MAR applies a different factorization as DART. Namely, MAR only partially denoise one patch at a time, while DART iteratively denoise the whole image which better leverages the global information. (2) MAR relies on an additional diffusion module alongside the AR model to denoise patches. Whereas DART organically integrates the denoising process into AR model through non-Markovian diffusion, which offers a more simplified and unified framework.
    - VAR [2] is an AR model that works on discrete tokens. It progressively predicts tokens of different resolutions and relies on a curated multi-resolution autoencoder tokenizer trained beforehand. Unlike VAR, our DART works on continuous latent space and doesn’t require curated multi-resolution tokens. Moreover, to generate at different resolutions, VAR requires re-train the autoencoder tokenizer while DART can adapts to higher-resolution generation easily by concatenating denoising steps of higher-resolution images (see Matryoshka-DART in point 4 below).
    - There are also families of vision-language model (VLM) like Chameleon [3] and Transfusion [4]. Chameleon models both text and image generate as autoregressive next-token prediction. And Transfusion trains a VLM but applying AR for text while applying diffusion for image generation separately. Unlike these models which develop multimodal generation in a brutal way, DART models image generation as autoregressive denoising which can be integrated seamlessly with AR text generation as shown in some preliminary results of Kaleido-DART. Besides, our model doesn’t focus on multimodal generation and just showcase the capability of text-image co-generation using our proposed framework. We believe DART shows the potential of building more unified and efficient multimodal generative models.

---

> ### Author Response · Authors · 2024-11-22
> **Official Response to All Reviewers (2)**
>
> 3. The reproduced results use the official codebase and checkpoint and we follow the best performing cfg scale reported in the original papers. We report reproduced results of DiT [5], SiT [6], and MAR [1] with 16 sampling times which is the same as our vanilla DART. Our model achieves competitive performance when compared with diffusion models like LDM [7] and AR models like VQGAN [8] and RQ-Transformer [9]. Admittedly, there is a gap between DART and SOTA visual generative models (like VAR [2] and MAR [1]). However, we want to point out that due to the limited computation resources, DART is trained with less FLOPs than many baselines. For example, DiT is trained for 7M iterations whereas DART is only trained for 500k iterations. Also, baselines like VAR and MAR employ larger models than our DART. In particular, VAR deploys a 2B model while the largest DART model is approximately 800M. Besides, in our reproduced results, when using only 16 sampling steps as the setting of our vanilla DART, our model show significantly better performance than DiT, SiT and MAR. Also, we report MAR-AR, a variant of MAR which generate tokens in an autoregressive manner instead of masked modeling which is applied in standard MAR models. DART which generates samples through autoregressive denoising shows better performance than MAR-AR. These results further validate the effectiveness of leveraging the whole denoising trajectory.
>     | Type | Model | FID | IS | #params | Steps |
>     | -------- | ------- | -------- | -------- | -------- | -------- |
>     | Diff. | ADM | 10.94 | 101.0 | 554M | 250 |
>     | | CDM | 4.88 | 158.7 | - | 8100 |
>     | | LDM | 3.60 | 247.7 | 400M | 250 |
>     | | DiT | 2.27 | 278.2 | 675M | 250 |
>     | | SiT |  2.06 | 277.5 | 675M | 250 |
>     | AR | VQGAN | 15.78 | 74.3 | 1.4B | 256 |
>     | | RQTran | 3.80 | 323.7 | 3.8B | 68 |
>     | | MAR-AR | 4.69 | 244.6 | 479M | 256 |
>     | | MAR | 1.55 | 303.7 | 943M | 256 |
>     | | VAR | 1.73 | 350.2 | 2.0B | 10 |
>     | Reprod. | DiT | 19.52 | 125.9 | 675M | 16 |
>     | | SiT* | 6.98 | 122.9 | 675M | 16 |
>     | | MAR* | 6.37 | 221.29 | 943M | 16 |
>     | Ours | DART | 5.62 | 231.7 | 812M | 16 |
>     | | DART-AR | 3.98 | 256.8 | 812M | 4096 |
>     | | DART-FM | 3.82 |  263.8 | 820M | 16 |
>
> 4. **DART can also be scaled to higher-resolution generation in an efficient way.** We have added the results of DART on image generation of resolutions $512 \times 512$ and $1024 \times 1024$ to Appendix D. In particular, we further extend DART to multi-resolution generation (i.e., Matryoshka-DART). Image patches of higher resolutions are concatenated after low resolution patches ($256 \times 256$) and the model progressive generates images from low to high resolutions. Figure 12 illustrates the process of progressively generating multi-resolution images. This paradigm also allows efficient generation for high-resolution images than standard diffusion model like DiT. Since the model can make full use of the generative trajectory of low-resolution samples, it can be directly finetuned from a pretrained $256 \times 256$ model by concatenating the higher-resolution tokens afterwards. Besides, the model requires less denoising steps for high-resolution generations since it is naturally conditioned on generated low-resolution images. Samples of high resolution results of resolutions $512 \times 512$ and $1024 \times 1024$ from text-to-image generation can be found in Figure 1, 14 and 15.

---

> > ### Author Response · Authors · 2024-11-22
> > **Official Response to All Reviewers (3)**
> >
> > References:
> >
> > [1] Li, Tianhong, Yonglong Tian, He Li, Mingyang Deng, and Kaiming He. "Autoregressive Image Generation without Vector Quantization." arXiv preprint arXiv:2406.11838 (2024).
> >
> > [2] Tian, Keyu, Yi Jiang, Zehuan Yuan, Bingyue Peng, and Liwei Wang. "Visual autoregressive modeling: Scalable image generation via next-scale prediction." arXiv preprint arXiv:2404.02905 (2024).
> >
> > [3] Team, Chameleon. "Chameleon: Mixed-modal early-fusion foundation models." arXiv preprint arXiv:2405.09818 (2024).
> >
> > [4] Zhou, Chunting, Lili Yu, Arun Babu, Kushal Tirumala, Michihiro Yasunaga, Leonid Shamis, Jacob Kahn, Xuezhe Ma, Luke Zettlemoyer, and Omer Levy. "Transfusion: Predict the next token and diffuse images with one multi-modal model." arXiv preprint arXiv:2408.11039 (2024).
> >
> > [5] Peebles, William, and Saining Xie. "Scalable diffusion models with transformers." In Proceedings of the IEEE/CVF International Conference on Computer Vision, pp. 4195-4205. 2023.
> >
> > [6] Ma, Nanye, Mark Goldstein, Michael S. Albergo, Nicholas M. Boffi, Eric Vanden-Eijnden, and Saining Xie. "Sit: Exploring flow and diffusion-based generative models with scalable interpolant transformers." arXiv preprint arXiv:2401.08740 (2024).
> >
> > [7] Rombach, Robin, Andreas Blattmann, Dominik Lorenz, Patrick Esser, and Björn Ommer. "High-resolution image synthesis with latent diffusion models." In Proceedings of the IEEE/CVF conference on computer vision and pattern recognition, pp. 10684-10695. 2022.
> >
> > [8] Esser, Patrick, Robin Rombach, and Bjorn Ommer. "Taming transformers for high-resolution image synthesis." In Proceedings of the IEEE/CVF conference on computer vision and pattern recognition, pp. 12873-12883. 2021.
> >
> > [9] Lee, Doyup, Chiheon Kim, Saehoon Kim, Minsu Cho, and Wook-Shin Han. "Autoregressive image generation using residual quantization." In Proceedings of the IEEE/CVF Conference on Computer Vision and Pattern Recognition, pp. 11523-11532. 2022.

---

### Meta-Review · Area_Chair_YCC1 · 2024-12-23

**Metareview:**

The paper proposes an interesting formulation using a non-Markovian diffusion model that uses autoregressive techniques to capture the full denoising trajectory, improving generation efficiency and flexibility. The paper also proposes DART-AR and DART-FM Modules: DART-AR enables token-level autoregressive modeling for enhanced control and quality, while DART-FM employs flow-based refinement to increase model expressiveness. Experimental results show promise in this formulation.

**Additional Comments On Reviewer Discussion:**

All reviewers agree that the idea proposed is interesting. One main concern raised by the reviewers is lack of experimental results and comparison with SOTA. The authors addressed these concerns in the rebuttal. One main issue though is the method doesn't outperform SOTA. But the method gives good performance on less number of steps, which is still a good result. While the paper is not SOTA, I still feel the contributions are good and would help future research. Hence, I vote for accepting the paper.

---

### Decision · Program_Chairs · 2025-01-22

Accept (Poster)